# Probabilistic Residual User Clustering

## Abstract

Modern recommender systems are typically based on deep learning (DL) models, where a dense encoder learns representations of users and items. As a result, these systems often suffer from the black-box nature and computational complexity of the underlying models, making it difficult to systematically interpret their outputs and enhance their recommendation capabilities. To address this problem, we propose *Probabilistic Residual User Clustering (PRUC)*, a causal Bayesian recommendation model based on user clustering. Specifically, we address this problem by (1) dividing users into clusters in an unsupervised manner and identifying causal confounders that influence latent variables, (2) developing sub-models for each confounder given the observable variables, and (3) generating recommendations by aggregating the rating residuals under each confounder using do-calculus. Experiments demonstrate that our *plug-and-play* PRUC is compatible with various base DL recommender systems, significantly improving their performance while automatically discovering meaningful user clusters.

## 1 Introduction

Over the past decade, personalized recommendations have significantly improved user experiences in domains such as e-commerce and social media. The recommender systems driving these advancements often rely on sophisticated deep learning (DL) models (Chung et al., 2014; Vaswani et al., 2017; Wu et al., 2019) capable of handling vast amounts of data, enabling highly accurate predictions and personalized interactions. Despite their effectiveness, these models often function as black boxes, lacking transparency and interpretability. This limitation poses significant challenges, particularly when diagnosing and enhancing the performance of recommender systems in scenarios involving domain shifts, such as changes in users' countries. Cold-start scenarios, a critical problem in recommendation systems, exacerbate these issues due to the presence of heterogeneous features and the influence of diverse and spurious patterns. As a result, existing models exhibit notably low performance in such settings.

Existing work (Yuan et al., 2020; Wu et al., 2020; Bi et al., 2020; Li et al., 2019; Hansen et al., 2020; Liang et al., 2020; Zhu et al., 2020; Liu et al., 2020; Kweon et al., 2024) often addresses domain shift by establishing connections across different domains through shared users or items. However, in real-world applications, such overlap is often unavailable. For instance, when recommending distinct items to users from different countries, there is typically no overlap in either users or items. This scenario demands more sophisticated modeling to account for shared confounders. For example, consider position/exposure bias in recommender systems: if the system ranks the item (e.g., an ad) higher, users are biased to rate it higher or have a higher probability to click it. Another example is popularity bias; users have a higher probability to click popular or trending items. A system must correct for such biases; otherwise, its accuracy will decline significantly when previously popular items lose their popularity. Additionally, existing methods often fail to consider latent user clusters when cluster IDs are not available in the datasets, therefore failing to model (dis)similarities among users.

To address these problems, we propose a novel causal hierarchical Bayesian deep learning model, dubbed *Probabilistic Residual User Clustering (PRUC)*, as a *plug-and-play* framework to improve and potentially interpret any base recommender systems in cross-domain settings. Fig. 1 shows the simplified overview of our framework. **During the training stage**, given any base recommender (e.g., DLRM (Naumov et al.,

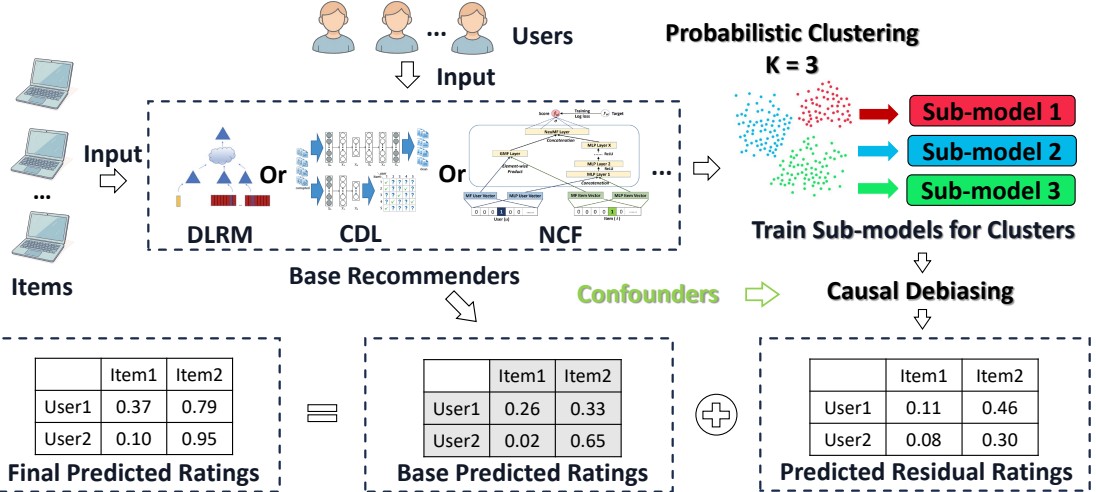

Figure 1: A detailed illustration of our proposed PRUC framework, highlighting the sequential stages and key components involved in its processing workflow. **Training:** Given *any base recommender* (e.g., DLRM (Naumov et al., 2019), CDL (Wang et al., 2015), or NCF (He et al., 2017)), PRUC is learned by jointly (1) computing its predicted ratings and the residual ratings (i.e., the difference between the ground-truth ratings and the base model's predicted ratings), (2) dividing users into latent clusters based on the residual ratings using our probabilistic clustering method, (3) training a sub-model for each latent user cluster. **Inference:** Once PRUC is learned, given a new user-item pair, PRUC can then (1) estimate the user's cluster ID to select the proper sub-model, (2) use this sub-model to perform causal inference to debias potential confounders and predict the residual rating, and (3) add this predicted residual rating to the base predicted rating to obtain the final predicted rating, thereby producing the final recommendation.

2019), CDL (Wang et al., 2015), or NCF (He et al., 2017)), PRUC is learned by jointly (1) computing its predicted ratings and the residual ratings (i.e., the difference between the ground-truth ratings and the base model's predicted ratings), (2) dividing users into latent clusters based on the residual ratings using our probabilistic clustering method, (3) training a sub-model for each latent user cluster. **During the inference stage**, once PRUC is learned, given a new user-item pair, PRUC can then (1) estimate the user's cluster ID to select the proper sub-model, (2) use this sub-model to perform causal inference to debias potential confounders and predict the residual rating, and (3) add this predicted residual rating to the base predicted rating to obtain the final predicted rating, thereby producing the final recommendation. Notably, PRUC is *plug-and-play*, i.e., it is compatible with any base DL recommendation model and can enhance the original model's performance.

Our contributions are as follows:

- We identify the existence of user clusters in various datasets, as well as latent confounders that have a causal effect on user and item hidden representations in DL models.

- We propose a causal Bayesian framework to discover the latent structures of users, items, and ratings. We incorporate user clusters and causal confounders as latent variables in the causal structural model (SCM) and perform inference via do-calculus over the confounders.

- We formulate the rating prediction problem as residual prediction, i.e., predicting the difference between the ground-truth user ratings and the base DL model's predicted ratings, to enhance the performance of base DL recommenders.

- Experiments verify that our *plug-and-play* PRUC is compatible with various base DL recommender systems, significantly improving their performance while automatically discovering meaningful user clusters.

## 2 Probabilistic Residual User Clustering

In this section, we describe our proposed PRUC framework.

### 2.1 Problem Setting and Notations

Consider a recommendation dataset containing $I$ users and $J$ items. A DL encoder $f_v(\cdot) : \mathbb{R}^d \to \mathbb{R}^h$ encodes each item $j$'s raw features $\mathbf{x}_j^v \in \mathbb{R}^d$ into $f_v(\mathbf{x}_j^v)$; assume there exists another decoder deep learning model $f_x(\cdot) : \mathbb{R}^h \to \mathbb{R}^d$, which decodes latent representation $\mathbf{v}_j$ back to the raw item features $\mathbf{x}_j^v$. For a given user $i$ and an item $j$, there is a ground-truth rating $R_{ij} \in \mathbb{R}$, a base predicted rating $\widehat{R}_{ij} \in \mathbb{R}$ provided by a base recommender, and a residual rating $\widetilde{R}_{ij} = R_{ij} - \widehat{R}_{ij}$. There is a latent cluster ID $k$ ($k \in \{1, ..., K\}$) that indicates which user group user $i$ belongs to. We assume that there exists a user latent vector $\mathbf{u}_i \in \mathbb{R}^h$ for each user $i$ and an item latent vector $\mathbf{v}_j \in \mathbb{R}^h$ for each item $j$; they are both impacted by a causal confounder $\mathbf{s} \in \mathbb{R}^g$, where $g \ll h$.

Our goal is to predict the final rating $R$ using the residual $R$, i.e., $R = \widehat{R} + \widetilde{R}$, where $\widehat{R}$ represents the rating from the original (base) DL recommender. When the original recommender is provided, $\widehat{R}$ is fixed; therefore we only need to learn $\widetilde{R}$ in order to predict the final rating $R$. For generality, we assume $M$ domains, where $m_i$ and $m_j$ denote the domain ID of user $i$ and item $j$, respectively.

### 2.2 Method Overview

We use a variational Bayesian framework to learn the latent parameters. Fig. 2 illustrates the corresponding probabilistic graphical model (PGM).

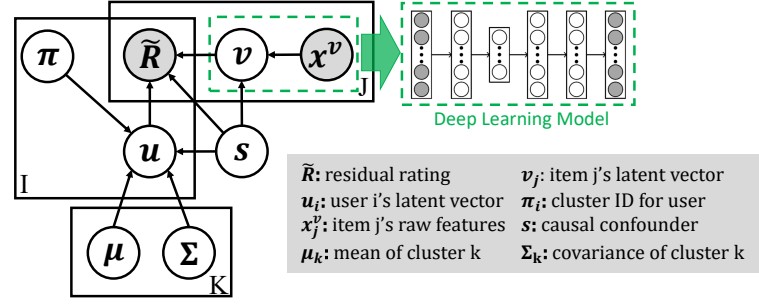

**Generative Process.** Below we describe the generative process of PRUC shown in Fig. 2.

For each domain $m \in \{1, 2, \ldots, M\}$:

- Draw the confounder $\mathbf{s}_m$ from a prior distribution, for example, $p(\mathbf{s}) \sim \mathcal{N}(\mathbf{0}, \mathbf{I})$:

Figure 2: Probabilistic graphical model of our PRUC framework.

- For each item $j$:
    - Draw item latent variable $v_j$ from the product of two Gaussians (Gales & Airey, 2006), i.e.,

$$\mathbf{v}_j \sim p(\mathbf{v}_j | \mathbf{x}_j^v, \mathbf{s}) = PoG(f_v(\mathbf{x}_j^v), \mathbf{W}^v \mathbf{s}^m, \Lambda_v^{-1}\mathbf{I}, \lambda_v^{-1}\mathbf{I}), \tag{1}$$

    where $\mathbf{W}^v$ is the learnable global parameter shared by all items, $f_v(\mathbf{x}_j^v)$ is a deep encoder, $\mathbf{I}$ is the identity matrix, and $\lambda_v \in \mathbb{R}$ and $\Lambda_v \in \mathbb{R}$ are the precision.

- For each user $i$:

    - Draw the user cluster ID $\pi_i$ from categorical distribution $p(\pi_i | \theta)$.
    - Draw user latent variable $\mathbf{u}_i$ from the $\pi_i$'th Gaussian distribution, i.e., $p(\mathbf{u}_i | \{\boldsymbol{\mu}_k, \boldsymbol{\Sigma}_k\}_{k=1}^K, \mathbf{s}, \pi) \sim \mathcal{N}(\boldsymbol{\mu}_{\pi_i} + \mathbf{W}^u \mathbf{s}^m, \boldsymbol{\Sigma}_{\pi_i})$.
      Notice that $\mathbf{W}^u$ is the learnable global parameter shared by all users.
    - For each item $j$:
        * Draw the residual rating $\widetilde{R}_{ij}$ from distribution $p(\widetilde{R}_{ij} | \mathbf{u}_i, \mathbf{v}_j, \mathbf{s}) \sim \mathcal{N}(\mathbf{u}_i^\top \mathbf{v}_j + \mathbf{w}^{R\top} \mathbf{s}_m, \lambda_{\widetilde{R}_{ij}}^{-1})$,

          where $\mathbf{w}^R$ is the learnable vector shared by all ratings and $\lambda_{\widetilde{R}_{ij}}$ is the precision.

$PoG(f_v(\mathbf{x}_j^v), \mathbf{W}^v \mathbf{s}^m, \Lambda_v^{-1}\mathbf{I}, \lambda_v^{-1}\mathbf{I})$ in Eqn. 1 denotes the product of two Gaussians, $\mathcal{N}(f_v(\mathbf{x}_j^v), \Lambda_v^{-1}\mathbf{I})$ and $\mathcal{N}(\mathbf{W}^v \mathbf{s}^m, \lambda_v^{-1}\mathbf{I})$, which is also a Gaussian (Gales & Airey, 2006), i.e., $\mathcal{N}(\boldsymbol{\mu}_{pog}, \lambda_{pog}\mathbf{I})$, where

$$\boldsymbol{\mu}_{pog} = \frac{\Lambda_v f_v(\mathbf{x}_j^v) + \lambda_v \mathbf{W}^v \mathbf{s}^m}{\Lambda_v + \lambda_v}, \qquad \lambda_{pog} = \Lambda_v + \lambda_v. \tag{2}$$

**Model Factorization**. As shown in Fig. 2, we factorize the generative model into four conditional distributions:

$$p(\mathbf{u}_i, \mathbf{v}_j, \pi, \widetilde{R}_{ij} | \{\boldsymbol{\mu}_k, \boldsymbol{\Sigma}_k\}_{k=1}^K, \mathbf{x}_j^v, \mathbf{s}_m)$$
$$= p(\widetilde{R}_{ij} | \mathbf{u}_i, \mathbf{v}_j, \mathbf{s}_m) p(\mathbf{u}_i | \{\boldsymbol{\mu}_k, \boldsymbol{\Sigma}_k\}_{k=1}^K, \mathbf{s}_m, \pi) p(\mathbf{v}_j | \mathbf{x}_j^v, \mathbf{s}_m) p(\pi). \tag{3}$$

$p(\pi|\theta)$ is the prior distribution for $\pi$. Each of the remaining distributions is assumed as a Gaussian distribution and is shown as follows:

$$p(\widetilde{R}_{ij} | \mathbf{u}_i, \mathbf{v}_j, \mathbf{s}_m) = \mathcal{N}(\mathbf{u}_i^\top \mathbf{v}_j + \mathbf{w}^{R\top} \mathbf{s}_m, \lambda_{\widetilde{R}_{ij}}^{-1}), \tag{4}$$

$$p(\mathbf{u}_i | \{\boldsymbol{\mu}_k, \boldsymbol{\Sigma}_k\}_{k=1}^K, \mathbf{s}_m, \pi) = \mathcal{N}(\boldsymbol{\mu}_{\pi_i} + \mathbf{W}^u \mathbf{s}_m, \boldsymbol{\Sigma}_{\pi_i}), \tag{5}$$

$$p(\mathbf{v}_j | \mathbf{x}_j^v, \mathbf{s}) = PoG(f_v(\mathbf{x}_j^v), \mathbf{W}^v \mathbf{s}^m, \Lambda_v^{-1}\mathbf{I}, \lambda_v^{-1}\mathbf{I}), \tag{6}$$

where $i$ and $j$ refers to the user index and the item index, respectively. We employ variational distributions $q(\mathbf{u}_i, \mathbf{v}_j | \mathbf{x}_j^v)$ to approximate the posterior distributions of $\mathbf{u}_i$ and $\mathbf{v}_i$.

$$q(\{\mathbf{u}_i\}_{i=1}^I, \{\mathbf{v}_j\}_{j=1}^J) = \prod_{i=1}^I q(\mathbf{u}_i) \prod_{j=1}^J q(\mathbf{v}_j). \tag{7}$$

More specifically, we assumes $q(\mathbf{v}_j)$ follows a gaussian distribution:

$$q(\mathbf{v}_j) = \mathcal{N}(\boldsymbol{\mu}_{\mathbf{v}_j}, \Lambda_v^{-1}\mathbf{I}). \tag{8}$$

Here, $j$ is the item index, $\Lambda_v \in \mathbb{R}$ refers to the precision. Similarly:

$$q(\mathbf{u}_i) = \mathcal{N}(\boldsymbol{\mu}_{\mathbf{u}_i}, \Lambda_u^{-1}\mathbf{I}), \tag{9}$$

where $i$ is the user index, and $\Lambda_u \in \mathbb{R}$ is the precision. Different users and items have different approximate posteriors. We also use a categorical variational distribution $q(\pi)$ to approximate the posterior distribution of $\pi$ (more details below).

**Learning Objective.** We maximize an evidence lower bound (ELBO) as our learning objective for both generative and inference model.

$$\mathcal{L}_{ELBO}(\mathbf{x}_j^v, \widetilde{R}_{ij})$$
$$= \mathbb{E}_{q(\mathbf{u}_i)q(\mathbf{v}_j)} \big[ \log p(\mathbf{u}_i, \mathbf{v}_j, \widetilde{R}_{ij} | \{\boldsymbol{\mu}_k, \boldsymbol{\Sigma}_k\}_{k=1}^K, \mathbf{x}_j^v, \mathbf{s}_m, \pi) \big]$$
$$+ \mathbb{E}_{q(\pi)}[p(\pi|\theta)] - \mathbb{E}_{q(\pi)}[q(\pi)]$$
$$- \mathbb{E}_{q(\mathbf{u}_i)q(\mathbf{v}_j)} \big[ \log q(\mathbf{v}_j) \big]$$
$$- \mathbb{E}_{q(\mathbf{u}_i)q(\mathbf{v}_j)} \big[ \log q(\mathbf{u}_i) \big]. \tag{10}$$

Combining Eqn. 3 and Eqn. 7, we obtain the following decomposition:

$$\mathcal{L}_{ELBO}(\mathbf{x}_j^v, \widetilde{R}_{ij})$$
$$= -D_{KL}\big(q(\mathbf{u}_i) \| p(\mathbf{u}_i | \{\boldsymbol{\mu}_k, \boldsymbol{\Sigma}_k\}_{k=1}^K, \mathbf{s}_m, \pi)\big) \tag{11}$$
$$+ \mathbb{E}_{q(\mathbf{u}_i)q(\mathbf{v}_j)q(\pi)} \big[ \log p(\widetilde{R}_{ij} | \mathbf{u}_i, \mathbf{v}_j, \mathbf{s}_m, \pi) \big] \tag{12}$$
$$- D_{KL}\big(q(\mathbf{v}_j) \| p(\mathbf{v}_j | \mathbf{x}_j^v, \mathbf{s}_m)\big), \tag{13}$$
$$- D_{KL}\big(q(\pi) \| p(\pi|\theta)\big), \tag{14}$$

where $D_{KL}(\cdot\|\cdot)$ is the Kullback-Leibler (KL) divergence. For Eqn. 11, we compute the log likelihood for each cluster $k$ as

$$\log p(\{\mathbf{u}_i\}_{i\in I_k}|\{\boldsymbol{\mu}_k, \boldsymbol{\Sigma}_k\}, \mathbf{s}_m, \pi) = -\frac{1}{2}\sum_{i\in I_k}[\log|\boldsymbol{\Sigma}_k| + (\mathbf{u}_i - \boldsymbol{\mu}_k - \mathbf{W}^u\mathbf{s}_m)^\top \boldsymbol{\Sigma}_k^{-1}(\mathbf{u}_i - \boldsymbol{\mu}_k - \mathbf{W}^u\mathbf{s}_m)] + C,$$

where $i$ is the user index, $I_k$ is the set of user index that belongs to cluster $k$, and $C$ is a constant.

Similarly, all the other terms can be expanded as:

$$\log p(\widetilde{R}_{ij}|\mathbf{u}_i, \mathbf{v}_j, \mathbf{s}) = -\frac{\lambda_{\widetilde{R}_{ij}}}{2}\left(\widetilde{R}_{ij} - \mathbf{u}_i^\top\mathbf{v}_j - \mathbf{w}^{R\top}\mathbf{s}_m\right)^2 + C, \tag{15}$$

$$D_{KL}\left(q(\mathbf{v}_j)\|p(\mathbf{v}_j|\mathbf{x}_j^v, \mathbf{s}_m)\right) = -\frac{\lambda_v}{2}\|\boldsymbol{\mu}_{\mathbf{v}_j} - \mathbf{W}^v\mathbf{s}_m\|^2$$
$$-\frac{\Lambda_v}{2}\|\boldsymbol{\mu}_{\mathbf{v}_j} - f_v(\mathbf{x}_j^v)\|^2 + C. \tag{16}$$

Here $C$ can be omitted because we treat scalars like $\lambda_v \in \mathbb{R}$, $\lambda_u \in \mathbb{R}$, $\Lambda_v \in \mathbb{R}$, and $\Lambda_u \in \mathbb{R}$ as constants.

**Intuition for Each Term in Eqn. 10.** Below, we describe the intuition of each term in Eqn. 10:

1. **Regularize Latent Variable $\mathbf{u}_i$ (Eqn. 11).** $D_{KL}\left(q(\mathbf{u}_i)\|p(\mathbf{u}_i|\{\boldsymbol{\mu}_k, \boldsymbol{\Sigma}_k\}_{k=1}^K, \mathbf{s}_m, \pi)\right)$ aims to regularize user $i$'s latent variable $\mathbf{u}_i$, ensuring $\mathbf{u}_i$ is close to the center of its corresponding user cluster $\pi_i$, and therefore close to other users' latent embeddings in the same cluster.

2. **Predict Residual Rating $\widetilde{R}_{ij}$ from $\mathbf{u}_i$ and $\mathbf{v}_j$ (Eqn. 12).** $p(\widetilde{R}_{ij}|\mathbf{u}_i, \mathbf{v}_j, \mathbf{s}_m)$ use the inferred $\mathbf{u}_i$, $\mathbf{v}_j$, and the causal confounder $\mathbf{s}_m$ to predict the residual rating, thereby encouraging $\mathbf{u}_i$ and $\mathbf{v}_j$ to retain more information to maximize prediction performance.

3. **Regularize Latent Variable $\mathbf{v}_j$ (Eqn. 13).** $D_{KL}(q(\mathbf{v}_j)\|p(\mathbf{v}_j|\mathbf{x}_j^v, \mathbf{s}_m))$ is the KL divergence term between the inference model $q(\cdot|\mathbf{x}_j^v)$ and the generative model $p(\cdot|\mathbf{s}_m)$; this encourages the inferred posterior $q(\mathbf{v}_j|\mathbf{x}_j^v)$ to be close to the prior distribution $p(\mathbf{v}_j|\mathbf{s}_m)$.

4. **Regularize Latent Variable $\pi$ (Eqn. 14).** $D_{KL}\left(q(\pi)\|p(\pi|\theta)\right)$ is the KL divergence term between the categorical variational distribution $q(\pi)$ and the prior $p(\pi|\theta)$; this encourages the inferred posterior $q(\pi)$ to be close to the prior $p(\pi|\theta)$.

### 2.3 Inference and Learning

In our framework, we need to learn several parameters, including the Gaussian parameters $\{\boldsymbol{\mu}_k, \boldsymbol{\Sigma}_k\}_{k=1}^K$, user latent $\mathbf{u}$, item latent $\mathbf{v}$, and the parameters of the functions $f_x(\cdot)$ and $f_v(\cdot)$, as well as $\mathbf{W}^u$, $\mathbf{W}^v$, and $\mathbf{w}^R$. The following sections detail the learning process for all these parameters. The complete algorithm is outlined in Algorithm 1.

**1) $\{\boldsymbol{\mu}_k, \boldsymbol{\Sigma}_k\}_{k=1}^K$.** To optimize $\{\boldsymbol{\mu}_k, \boldsymbol{\Sigma}_k\}_{k=1}^K$, we take derivative of Eqn. 15 w.r.t. $\boldsymbol{\mu}_k$ and $\boldsymbol{\Sigma}_k$ as follows:

$$\frac{\partial\mathcal{L}}{\partial\boldsymbol{\mu}_k} = \boldsymbol{\Sigma}_k^{-1}\left(\mathbf{u}_i - \boldsymbol{\mu}_k - \mathbf{W}^u\mathbf{s}_m\right), \tag{17}$$

$$\frac{\partial\mathcal{L}}{\partial\boldsymbol{\Sigma}_k} = \frac{1}{2}\boldsymbol{\Sigma}_k^{-1}\left[\left(\mathbf{u}_i - \boldsymbol{\mu}_k - \mathbf{W}^u\mathbf{s}_m\right)\left(\mathbf{u}_i - \boldsymbol{\mu}_k - \mathbf{W}^u\mathbf{s}_m\right)^\top - \boldsymbol{\Sigma}_k\right]\boldsymbol{\Sigma}_k^{-1}. \tag{18}$$

Setting Eqn. 17 and Eqn. 18 to zero leads to the following update rules, respectively:

$$\boldsymbol{\mu}_k = \frac{1}{|I_k|}\sum_{i\in I_k}\left(\mathbf{u}_i - \mathbf{W}^u\mathbf{s}_m\right), \tag{19}$$

$$\boldsymbol{\Sigma}_k = \frac{1}{|I_k|}\sum_{i\in I_k}\left(\mathbf{u}_i - \boldsymbol{\mu}_k - \mathbf{W}^u\mathbf{s}_m\right)\left(\mathbf{u}_i - \boldsymbol{\mu}_k - \mathbf{W}^u\mathbf{s}_m\right)^\top, \tag{20}$$

---

**Algorithm 1** Learning Algorithm of PRUC

---

**Input:** Raw item features $\mathbf{x}^v$, initialized $f_x(\cdot)$ and $f_v(\cdot)$ parameters, $\mathbf{W}^u, \mathbf{W}^v, \mathbf{w}^R$, initialized Gaussian parameters $\{\boldsymbol{\mu}_k, \boldsymbol{\Sigma}_k\}_{k=1}^K$, and the number of epochs T.
**for** $t = 1 : T$ **do**
  **for** $m = 1 : M$ **do**
    Update $\mathbf{u}_i$ and $\mathbf{v}_j$ using Eqn. 21 and Eqn. 22 update the variational distribution of $\pi_i$ using Eqn. 23.
    Update $\mathbf{W}^u, \mathbf{W}^v, \mathbf{w}^R$ using Eqn. 24, Eqn. 25 and Eqn. 26.
    Update the parameters of $f_v(\cdot)$ using gradient ascent of $\mathcal{L}$ in Eqn. 10.
  Update $\{\boldsymbol{\mu}_k, \boldsymbol{\Sigma}_k\}_{k=1}^K$ using Eqn. 19 and Eqn. 20, respectively; update parameters of $f_x(\cdot)$ using gradient ascent of $\mathcal{L}$ in Eqn. 10.
**Output:** $f_x(\cdot)$ and $f_v(\cdot)$ parameters, $\mathbf{W}^u, \mathbf{W}^v, \mathbf{w}^R$, and Gaussian parameters $\{\boldsymbol{\mu}_k, \boldsymbol{\Sigma}_k\}_{k=1}^K$.

---

where $\mathbf{I}_k$ is the set of user index $i$ that belongs to cluster $k$.

**2) $\mathbf{u}_i, \mathbf{v}_j$, and $\pi_i$.** After computing the gradients of Eqn. 10 w.r.t. to the means of $\mathbf{u}_i \sim \mathcal{N}(\boldsymbol{\mu}_{\mathbf{u}_i}, \Lambda_u^{-1}\mathbf{I})$ (i.e., $\boldsymbol{\mu}_{\mathbf{u}_i}$) and $\mathbf{v}_j \sim \mathcal{N}(\boldsymbol{\mu}_{\mathbf{v}_j}, \Lambda_v^{-1}\mathbf{I})$ (i.e., $\boldsymbol{\mu}_{\mathbf{v}_j}$), we obtain the following update rules:

$$\boldsymbol{\mu}_{\mathbf{u}_i} = (\boldsymbol{\Sigma}_{\pi_i}\mathbf{V}\lambda_{\widetilde{R}_{(i,:)}}\mathbf{V}^\top + \mathbf{I})^{-1}\left[\boldsymbol{\mu}_{\pi_i} + \mathbf{W}^u\mathbf{s}_m + \boldsymbol{\Sigma}_{\pi_i}\mathbf{V}\lambda_{\widetilde{R}_{(i,:)}}(\widetilde{\mathbf{R}}_{(i,:)} - \mathbf{w}^{R\top}\mathbf{s}_m\mathbf{I})\right], \tag{21}$$

$$\boldsymbol{\mu}_{\mathbf{v}_j} = \left[\mathbf{U}\lambda_{\widetilde{R}_{(:,j)}}\mathbf{U}^\top + (\lambda_v - \Lambda_v)\mathbf{I}\right]^{-1}\left[\lambda_v\mathbf{W}^v\mathbf{s}_m - \Lambda_v f_v(\mathbf{x}_j^v) + \mathbf{U}\lambda_{\widetilde{R}_{(:,j)}}(\widetilde{\mathbf{R}}_{(:,j)} - \mathbf{w}^{R\top}\mathbf{s}_m\mathbf{I})\right]. \tag{22}$$

Note that here $\mathbf{U}$ and $\mathbf{V}$ refer to user latent matrix $(\mathbf{u}_i)_{i=1}^I$ and item latent matrix $(\mathbf{v}_j)_{j=1}^J$. $\widetilde{\mathbf{R}}_{(i,:)} := (\widetilde{R}_{i1}, \cdots, \widetilde{R}_{iJ})^\top$, $\widetilde{\mathbf{R}}_{(:,j)} := (\widetilde{R}_{1j}, \cdots, \widetilde{R}_{Ij})^\top$. $\lambda_{\widetilde{R}_{(i,:)}} := \text{diag}(\lambda_{\widetilde{R}_{i1}}, \cdots, \lambda_{\widetilde{R}_{iJ}})$, $\lambda_{\widetilde{R}_{(:,j)}} := \text{diag}(\lambda_{\widetilde{R}_{1j}}, \cdots, \lambda_{\widetilde{R}_{Ij}})$.

With a uniform prior distribution on $\pi_i$, i.e., $p(\pi_i = k|\theta) = \frac{1}{K}$, the estimation of $\pi_i$ is similar to that of a Gaussian mixture model (GMM). Specifically, we can approximate $p(\pi_i = k|u_i, v_j, x_j^v, \{\mu_k, \Sigma_k\}_{k=1}^K)$ using

$$q(\pi_i = k) = \frac{\mathcal{N}(\mathbf{u}_i; \mu_k + W_u s_{m_i}, \Sigma_k)}{\sum_{l=1}^K \mathcal{N}(\mathbf{u}_i; \mu_l + W_u s_{m_i}, \Sigma_l)}, \tag{23}$$

which is obtained by maximizing the ELBO Eqn. 11~14. We then choose the optimal (most probable) cluster for each user $i$, i.e., the cluster assignment for user $i$ is set to $\pi_i = \text{argmax}_k q(\pi_i = k)$, which is used to compute Eqn. 21~22.

**3) $\mathbf{W}^u, \mathbf{W}^v, \mathbf{w}^R$.** The update rules for $\mathbf{W}^u$, $\mathbf{W}^v$, and $\mathbf{w}^R$ are as follows:

$$\mathbf{W}^u = \frac{1}{I}(\sum_{i=1}^I \mathbf{u}_i - \sum_{k=1}^K |I_k|\boldsymbol{\mu}_k)\mathbf{s}_m^\top(\mathbf{s}_m\mathbf{s}_m^\top)^{-1}, \tag{24}$$

$$\mathbf{W}^v = \frac{1}{J}\sum_{j=1}^J \mathbf{v}_j\mathbf{s}_m^\top(\mathbf{s}_m\mathbf{s}_m^\top)^{-1}, \tag{25}$$

$$\mathbf{w}^R = \frac{\sum_{i,j}\lambda_{\widetilde{R}_{ij}}(\widetilde{R}_{ij} - \mathbf{u}_i^\top\mathbf{v}_j)}{\sum_{i,j}\lambda_{\widetilde{R}_{ij}}}(\mathbf{s}_m\mathbf{s}_m^\top)^{-1}\mathbf{s}_m. \tag{26}$$

**4) Parameters of $f_x(\cdot)$ and $f_v(\cdot)$.** We use gradient ascent of $\mathcal{L}$ in Eqn. 10 to update these parameters.

**Inference.** Inference includes the *E-Step* in Algorithm 1, where PRUC infers latent variables $\mathbf{u}_i$ and $\mathbf{v}_j$, and updates the parameters of encoder model $f_v(\cdot)$ using gradient ascent of $\mathcal{L}$ in Eqn. 10.

**Learning.** Learning includes the iteration between the *E-Step* and *M-Step* in Algorithm 1 until convergence. In each *M-Step*, we update the Gaussian parameters $\{\boldsymbol{\mu}_k, \boldsymbol{\Sigma}_k\}_{k=1}^K$ following the update rules from Eqn. 19 and Eqn. 20, respectively; we update learnable parameters $\mathbf{W}^u, \mathbf{W}^v, \mathbf{w}^R$ following the update rules from Eqn. 24, Eqn. 25, and Eqn. 26, respectively. We also update parameters of the decoder $f_x(\cdot)$ with a reconstruction loss, $\||f_x(f_v(\mathbf{x}_j^v)) - \mathbf{x}_j^v|\|_2^2$ to regularize the encoding $f_v(\mathbf{x}_j^v)$ from the encoder $f_v(\cdot)$.

### 2.4 Plug-and-Play PRUC

Below we discuss key components of our plug-and-play PRUC after learning all parameters with Algorithm 1.

**Inferring User Cluster $\pi_i$.** With the learned Gaussian mixture's parameters, i.e., the mean and covariance $\boldsymbol{\mu}_k$ and $\boldsymbol{\Sigma}_k$ for each Gaussian component $k$ (each *Gaussian component* represents one *user cluster*) from Algorithm 1, PRUC infers the cluster for each user $i$, i.e., $p(\pi_i|\widetilde{R}_{ij}, \{\mathbf{u}_i\}, \{\mathbf{v}_j\}, \{\boldsymbol{\mu}_k, \boldsymbol{\Sigma}_k\}_{k=1}^K)$, determining which cluster $\pi_i$ user $i$ belongs to.

**Isolating Causal Confounders $\mathbf{s}_m$.** With the structured causal model (SCM), we isolate the *causal confounders* $\mathbf{s}_m$ for each domain $m$ by approximating its posterior distribution $p(\mathbf{s}_m|\widetilde{R}, \mathbf{x}_j^v, \{\boldsymbol{\mu}_k, \boldsymbol{\Sigma}_k\}_{k=1}^K)$ via variational domain indexing (VDI) (Xu et al., 2023). In this way, we can minimize the bias introduced by the causal confounder $\mathbf{s}_m$ when inferring $\mathbf{u}_i$ and $\mathbf{v}_j$ using Eqn. 21 and Eqn. 22, respectively. The domain index $s_m$ can be thought of as the embedding for each domain $m$. For example, in the dataset XMRec where each of the 18 domains contains items and users from one market/country (e.g., France or US), $s_m$ can be thought of as a "country" embedding. Interestingly, our preliminary results show that similar countries tend to have similar domain embedding $s_m$ (i.e., domain index) (Xu et al., 2023) . In other words, $s_m$ captures the similarities among different domains and therefore provide valuable information for our recommender systems.

**Debiasing the Causal Confounders.** Under our *PRUC* framework, for each inferred user cluster $k$, we perform causal inference for each user $i$ in this cluster to predict the residual $\widetilde{R}_{ij}$ (for each item $j$) while debiasing the causal confounders $\mathbf{s}$. Specifically, with the inferred $\mathbf{u}_i$ and $\mathbf{v}_j$ (using Eqn. 21 and Eqn. 22) and $\mathbf{s}_m$, we can predict $\widetilde{R}_{ij}$ by do-calculus as

$$p^{(k)}(\widetilde{R}_{ij}|do(\mathbf{u}_i), do(\mathbf{v}_j)) = \sum_{m=1}^M p^{(k)}(\widetilde{R}_{ij}|\mathbf{u}_i, \mathbf{v}_j, \mathbf{s}_m)p(\mathbf{s}_m), \tag{27}$$

where $p^{(k)}(\widetilde{R}_{ij}|\mathbf{u}_i, \mathbf{v}_j, \mathbf{s})$ represents the $k$'th sub-model trained from the $k$'th cluster's user data. In practice, we use $k = \pi_i$ ($\pi_i$ is user $i$'s cluster) when predicting user $i$'s rating $\widetilde{R}_{ij}$.

Note that performing causal inference by intervening $(\mathbf{u}_i, \mathbf{v}_j)$ effectively cuts the relations between the causal confounders $\mathbf{s}$ and $(\mathbf{u}_i, \mathbf{v}_j)$. Fig. 3 demonstrate the do-calculus that PRUC performs for debiasing the causal confounder $\mathbf{s}$.

**Intuition behind Do-Calculus.** The rationale of performing do-calculus in PRUC is that getting interventional distributions often requires intervening the recommender system to collect training data, which is expensive in practice. In contrast, do-calculus works by leveraging existing data to estimate the conditional distribution $p^{(k)}(\widetilde{R}_{ij}|\mathbf{u}_i, \mathbf{v}_j, \mathbf{s})$, and therefore prevent the potential cost (and risk) of actually intervening the system. Moreover, in datasets like XMRec, where each domain is composed of users and items from a single country, the "country" variable (denoted by its embedding $s_m$ in Section 2.4) can function as a confounder, introducing biases such as exposure

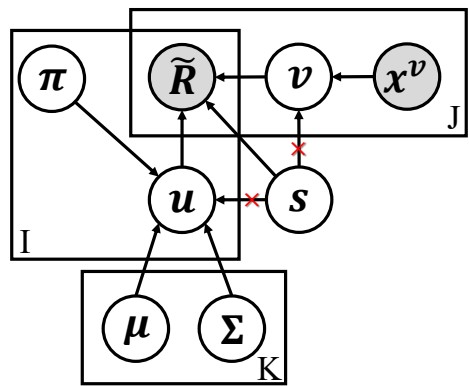

Figure 3: Causal inference in PRUC is equivalent to cutting the the confounder $\mathbf{s}$'s influence on $\mathbf{u}_i$ and $\mathbf{v}$.

bias and spurious correlations. This arises because the model may inadvertently attribute user-item preferences to country-specific factors, thereby distorting the true causal relationships between users and items and leading to misleading inferences. For example, a model might learn that "users who like Bollywood films also like cricket gear," not because of a true shared interest, but because both items are popular in India. In such cases, do-calculus becomes crucial, as it allows us to perform causal inference to control for such confounders, ensuring that the model's predictions are based on true relationships rather than spurious associations.

**Summary.** To summarize, for each user $i$, PRUC causally infer the residual rating $\widetilde{R}_i$ as follows:

1. Infer the user cluster $\pi_i$ by approximating its posterior $p(\pi_i|\mathbf{u}_i, \mathbf{v}_j, \mathbf{x}_j^v, \{\boldsymbol{\mu}_k, \boldsymbol{\Sigma}_k\}_{k=1}^K)$.

Table 1: Performance of PRUC with different base models on XMRec. The best results are marked with **bold face**.

| Data | Method | Recall@20 | F1@20 | MAP@20 | NDCG@20 | Precision@20 |
|---|---|---|---|---|---|---|
| France, Italy, India →Japan, Mexico | CDL (Base Model)
PRUC (Full) | 0.0143
**0.1091** | 0.0016
**0.0128** | 0.0028
**0.0463** | 0.0009
**0.0108** | 0.0009
**0.0068** |
| | DLRM (Base Model)
PRUC (Full) | 0.0044
**0.0295** | 0.0004
**0.0035** | 0.0004
**0.0048** | 0.0002
**0.0018** | 0.0002
**0.0018** |
| | PerK (Base Model)
PRUC (Full) | 0.1098
**0.1635** | 0.0128
**0.0192** | 0.0512
**0.0637** | 0.0112
**0.0151** | 0.0068
**0.0102** |
| | NCF (Base Model)
PRUC (Full) | 0.0131
**0.1137** | 0.0148
**0.0137** | 0.0026
**0.0309** | 0.0008
**0.0090** | 0.0008
**0.0073** |
| | LightGCN (Base Model)
PRUC (Full) | 0.0182
**0.1003** | 0.0021
**0.0121** | 0.0050
**0.0316** | 0.0014
**0.0084** | 0.0011
**0.0064** |
| Mexico, Spain, India →Japan, Germany | CDL (Base Model)
PRUC (Full) | 0.1127
**0.1761** | 0.0135
**0.0230** | 0.0301
**0.0593** | 0.0086
**0.0163** | 0.0072
**0.0123** |
| | DLRM (Base Model)
PRUC (Full) | 0.0756
**0.2017** | 0.0093
**0.0246** | 0.0085
**0.0545** | 0.0041
**0.0156** | 0.0049
**0.0131** |
| | PerK (Base Model)
PRUC (Full) | 0.1443
**0.2750** | 0.0177
**0.0335** | 0.0601
**0.1086** | 0.0143
**0.0263** | 0.0094
**0.0179** |
| | NCF (Base Model)
PRUC (Full) | 0.0096
**0.1558** | 0.0012
**0.0202** | 0.0022
**0.0280** | 0.0007
**0.0107** | 0.0007
**0.0108** |
| | LightGCN (Base Model)
PRUC (Full) | 0.0165
**0.1064** | 0.0022
**0.0138** | 0.0061
**0.0278** | 0.0016
**0.0087** | 0.0012
**0.0077** |
| Germany, Italy, Japan →United States, India | CDL (Base Model)
PRUC (Full) | 0.0252
**0.0257** | 0.0055
**0.0058** | **0.0084**
0.0078 | 0.0040
**0.0041** | 0.0031
**0.0033** |
| | DLRM (Base Model)
PRUC (Full) | 0.0024
**0.0066** | 0.0006
**0.0016** | 0.0003
**0.0024** | 0.0003
**0.0012** | 0.0003
**0.0009** |
| | PerK (Base Model)
PRUC (Full) | 0.0148
**0.0207** | 0.0033
**0.0046** | 0.0041
**0.0060** | 0.0022
**0.0031** | 0.0018
**0.0026** |
| | NCF (Base Model)
PRUC (Full) | 0.0018
**0.0126** | 0.0005
**0.0033** | 0.0004
**0.0021** | 0.0003
**0.0018** | 0.0003
**0.0019** |
| | LightGCN (Base Model)
PRUC (Full) | 0.0016
**0.0052** | 0.0004
**0.0013** | 0.0002
**0.0013** | 0.0002
**0.0008** | 0.0003
**0.0007** |

2. Infer the residual rating $\widetilde{R}_{ij}$ by causal Bayesian model averaging defined in Eqn. 27.

3. Predict the final rating as $R = \widetilde{R} + \widehat{R}$, where $\widehat{R}$ is the base recommender's prediction.

## 3 Experiments

In this section, we evaluate our PRUC as a plug-and-play framework to enhance arbitrary base recommenders on *XMRec* and *MovieLens*.

### 3.1 Datasets

**XMRec.** *XMRec* (Bonab et al., 2021) is a dataset encompassing 18 local markets (i.e., countries), 16 distinct product categories, and 52.5 million user-item interactions. For each item $j$, we use its item descriptions from the dataset as the item features $\mathbf{x}_j^v$. Users with fewer than three purchases are excluded from experiments. We use three training-testing domain splits: France, Italy, India → Japan, Mexico; Mexico, Spain, India → Japan, Germany; and Germany, Italy, Japan → United States, India. We use the production country of the products as the casual confounders $\mathbf{s}_m$.

**MovieLens.** *MovieLens* (Harper & Konstan, 2015) features movie ratings from users of varying ages. We use movie titles and movie plots as the item features $\mathbf{x}_j^v$. User features are derived from the first three films each user rated. Users who rated fewer than five movies or whose ratings do not exceed 3 are omitted. Post-filtering, our experiments involve 6,034 users and 3,705 items. We use two training-testing domain splits based on user ages: 1-18, 18-25, 35-45, 45-50, 50-56, $56^+$ → 25-35; and 25-35 → all the previous mentioned age groups. For brevity, we refer to each age group by the starting age, e.g., "1" for "1-18". We use the normalized movie released years as causal confounders $\mathbf{s}_m$.

Table 2: Performance of PRUC with different base models on MovieLens. The best results are marked with **bold face**.

| Data | Method | Recall@20 | F1@20 | MAP@20 | NDCG@20 | Precision@20 |
|---|---|---|---|---|---|---|
| 1, 18, 35, 45, 50, 56 →25 | CDL (Base Model) | 0.0179 | 0.0274 | 0.0045 | 0.0581 | 0.0587 |
| | PRUC (Full) | **0.0252** | **0.0409** | **0.0072** | **0.1071** | **0.1076** |
| | DLRM (Base Model) | 0.0714 | 0.1096 | **0.0285** | **0.2433** | 0.2366 |
| | PRUC (Full) | **0.0716** | **0.1101** | 0.0284 | 0.2431 | **0.2372** |
| | PerK (Base Model) | 0.0682 | 0.1029 | **0.0290** | **0.2224** | 0.2107 |
| | PRUC (Full) | **0.0690** | **0.1037** | 0.0287 | 0.2190 | **0.2110** |
| | NCF (Base Model) | 0.0050 | 0.0250 | 0.0011 | 0.0251 | 0.0251 |
| | PRUC (Full) | **0.0240** | **0.0387** | **0.0057** | **0.0947** | **0.1005** |
| | LightGCN (Base Model) | 0.0081 | 0.0132 | 0.0019 | 0.0381 | 0.0358 |
| | PRUC (Full) | **0.0249** | **0.0402** | **0.0069** | **0.1076** | **0.1055** |
| 25 →1, 18, 35, 45, 50, 56 | CDL (Base Model) | 0.0576 | 0.0848 | 0.0174 | 0.1602 | 0.1716 |
| | PRUC (Full) | **0.0645** | **0.0952** | **0.0202** | **0.1772** | **0.1897** |
| | DLRM (Base Model) | 0.0848 | 0.1342 | 0.0382 | 0.3347 | 0.3225 |
| | PRUC (Full) | **0.0903** | **0.1405** | **0.0414** | **0.3455** | **0.3319** |
| | PerK (Base Model) | 0.0746 | 0.1164 | 0.0324 | 0.2701 | 0.2661 |
| | PRUC (Full) | **0.0792** | **0.1225** | **0.0355** | **0.2821** | **0.2757** |
| | NCF (Base Model) | 0.0140 | 0.0229 | 0.0030 | 0.0633 | 0.0652 |
| | PRUC (Full) | **0.0450** | **0.0694** | **0.0144** | **0.1639** | **0.1711** |
| | LightGCN (Base Model) | 0.0093 | 0.0157 | 0.0022 | 0.0497 | 0.0482 |
| | PRUC (Full) | **0.0290** | **0.0480** | **0.0097** | **0.1493** | **0.1395** |

In all experiments, we use a cold-start setting where each testing domain user has only one rating in the training set, making the recommendations extremely challenging.

## 3.2 Base Recommenders and Baselines

Note that our PRUC method is a *plug-and-play* solution, compatible with *any* base recommenders. In this paper, we select the following five base recommenders to demonstrate PRUC's enhancement of state-of-the-art recommendation models.

- **CDL** (Wang et al., 2015) is a Bayesian deep framework that jointly integrates deep representation learning of content information with collaborative filtering on the ratings (feedback) matrix within a unified model.
- **DLRM** (Naumov et al., 2019) learns embeddings to represent both sparse and dense features by a neural network and predicts event probability.
- **PerK** (Kweon et al., 2024) uses calibrated interaction probabilities to determine the expected user utility and selects the optimal recommendation size $K$ to maximize it.
- **NCF** (He et al., 2017) proposes a generalized matrix factorization framework by replacing the inner product with a trainable neural network.
- **LightGCN** (He et al., 2020) simplifies the design of Graph Convolutional Networks (GCNs) for recommendation tasks, making it easier to train and enhancing overall performance compared with traditional GCNs.

Here CDL, DLRM, PerK, NCF and LightGCN serve as both (1) our **baselines** to compare against and (2) our **base recommenders** to enhance (see Fig. 2). For more details on training configurations, see Appendex B.2.

## 3.3 Metrics

We use five metrics for evaluation: Recall, Precision, mAP, F1, and NDCG.

Table 3: Comparison between PRUC w/o Causality and PRUC (Full) on a specific domain with different base models on XMRec. The best results are marked with **bold face**.

| Data | Method | Recall@20 | F1@20 | MAP@20 | NDCG@20 | Precision@20 |
|---|---|---|---|---|---|---|
| France, Italy, India →Japan, Mexico | CDL (Base Model) | 0.0143 | 0.0016 | 0.0028 | 0.0009 | 0.0009 |
| | CDL PRUC w/o Causality | 0.1058 | 0.0126 | 0.0333 | 0.0088 | 0.0067 |
| | PRUC (Full) | **0.1091** | **0.0128** | **0.0463** | **0.0108** | **0.0068** |
| | DLRM (Base Model) | 0.0044 | 0.0004 | 0.0004 | 0.0002 | 0.0002 |
| | DLRM PRUC w/o Causality | 0.0232 | 0.0026 | 0.0039 | 0.0014 | 0.0014 |
| | PRUC (Full) | **0.0295** | **0.0035** | **0.0048** | **0.0018** | **0.0018** |
| | PerK (Base Model) | 0.1098 | 0.0128 | 0.0512 | 0.0112 | 0.0068 |
| | PerK PRUC w/o Causality | 0.1376 | 0.0160 | 0.0558 | 0.0129 | 0.0085 |
| | PRUC (Full) | **0.1635** | **0.0192** | **0.0637** | **0.0151** | **0.0102** |
| | NCF (Base Model) | 0.0131 | 0.0148 | 0.0026 | 0.0008 | 0.0008 |
| | NCF PRUC w/o Causality | 0.1056 | 0.0126 | 0.0235 | 0.0074 | 0.0067 |
| | PRUC (Full) | **0.1137** | **0.0137** | **0.0309** | **0.0090** | **0.0073** |
| | LightGCN (Base Model) | 0.0182 | 0.0021 | 0.0050 | 0.0014 | 0.0011 |
| | LightGCN PRUC w/o Causality | 0.0940 | 0.0112 | 0.0289 | 0.0076 | 0.0059 |
| | PRUC (Full) | **0.1003** | **0.0121** | **0.0316** | **0.0084** | **0.0064** |

**Recall.** Recall@$N$ measures the proportion of relevant items retrieved among the top $N$ recommended items for user $i$:

$$\text{Recall}_i@N = \sum_{n=1}^{N} \text{rel}_{i,n}/|J_i|, \tag{28}$$

where $\text{rel}_{i,n}$ is an indicator that equals 1 if the item at rank $n$ is relevant to user $i$, and 0 otherwise. $|J_i|$ denotes the total number of relevant items for user $i$.

**Precision.** Precision@$N$ measures the proportion of the top $N$ recommended items that are relevant to user $i$:

$$\text{Precision}_i@N = \sum_{n=1}^{N} \text{rel}_{i,n}/N, \tag{29}$$

where $\text{rel}_{i,n}$ is 1 if the item at rank $n$ is relevant to user $i$, and 0 otherwise.

**mAP.** Mean Average Precision (mAP) computes the average precision over all relevant items for user $i$. See Appendix B.1 for more details.

**F1-score.** The F1 Score@$N$ for user $i$ is the harmonic mean of Precision@$N$ and Recall@$N$, providing a balance between the two metrics:

$$\text{F1}_i@N = 2 \times {}^{\text{Precision}_i@N \times \text{Recall}_i@N}/_{\text{Precision}_i@N + \text{Recall}_i@N}, \tag{30}$$

where $\text{Recall}_i@N$ and $\text{Precision}_i@N$ are defined in Eqn. 28 and Eqn. 29, respectively.

**NDCG.** Normalized Discounted Cumulative Gain (NDCG@$N$) evaluates the quality of the ranked list by considering the positions of the relevant items, giving higher scores to items appearing earlier in the list. See Appendix B.1 for more details.

**All metrics are computed by averaging over all users $i$.**

### 3.4 Results

**Results for Different Base Models.** Table 1 and Table 2 show the performance of PRUC with various base models across different metrics on both datasets. Results show that our full model ("PRUC (Full)") can generally boosts the base models' performance.

**Recall@$N$ with Larger $N$.** Fig. 4 shows Recall@$N$ for $N = 50, 100, 150, 200, 250, 300$ across three base models (CDL, DLRM, and PerK) and three training-testing domain splits. These figures indicate that PRUC surpasses the base models even without the causality component ("PRUC w/o Causality"), while full PRUC consistently outperforms its non-causal counterpart in all settings. We observe similar results for other base models.

Table 4: Performance of PRUC with different base models on MovieLens. The best results are marked with **bold face**.

| Data | Method | Recall@20 | F1@20 | MAP@20 | NDCG@20 | Precision@20 |
|---|---|---|---|---|---|---|
| 1, 18, 35, 45, 50, 56 →25 | CDL (Base Model) | 0.0179 | 0.0274 | 0.0045 | 0.0581 | 0.0587 |
| | PRUC w/o Causality | 0.0186 | 0.0302 | 0.0057 | 0.0863 | 0.0801 |
| | PRUC (Full) | **0.0252** | **0.0409** | **0.0072** | **0.1071** | **0.1076** |
| | DLRM (Base Model) | 0.0714 | 0.1096 | **0.0285** | **0.2433** | 0.2366 |
| | PRUC w/o Causality | 0.0232 | 0.0026 | 0.0039 | 0.0014 | 0.0014 |
| | PRUC (Full) | **0.0716** | **0.1101** | 0.0284 | 0.2431 | **0.2372** |
| | PerK (Base Model) | 0.0682 | 0.1029 | **0.0290** | **0.2224** | 0.2107 |
| | PRUC w/o Causality | 0.0582 | 0.0877 | 0.0212 | 0.1755 | 0.1787 |
| | PRUC (Full) | **0.0690** | **0.1037** | 0.0287 | 0.2190 | **0.2110** |
| | NCF (Base Model) | 0.0050 | 0.0250 | 0.0011 | 0.0251 | 0.0251 |
| | PRUC w/o Causality | 0.0231 | 0.0374 | 0.0055 | 0.0927 | 0.0989 |
| | PRUC (Full) | **0.0240** | **0.0387** | **0.0057** | **0.0947** | **0.1005** |
| | LightGCN (Base Model) | 0.0081 | 0.0132 | 0.0019 | 0.0381 | 0.0358 |
| | PRUC w/o Causality | 0.0248 | 0.0402 | 0.0070 | 0.1077 | 0.1053 |
| | PRUC (Full) | **0.0249** | **0.0402** | **0.0069** | **0.1076** | **0.1055** |

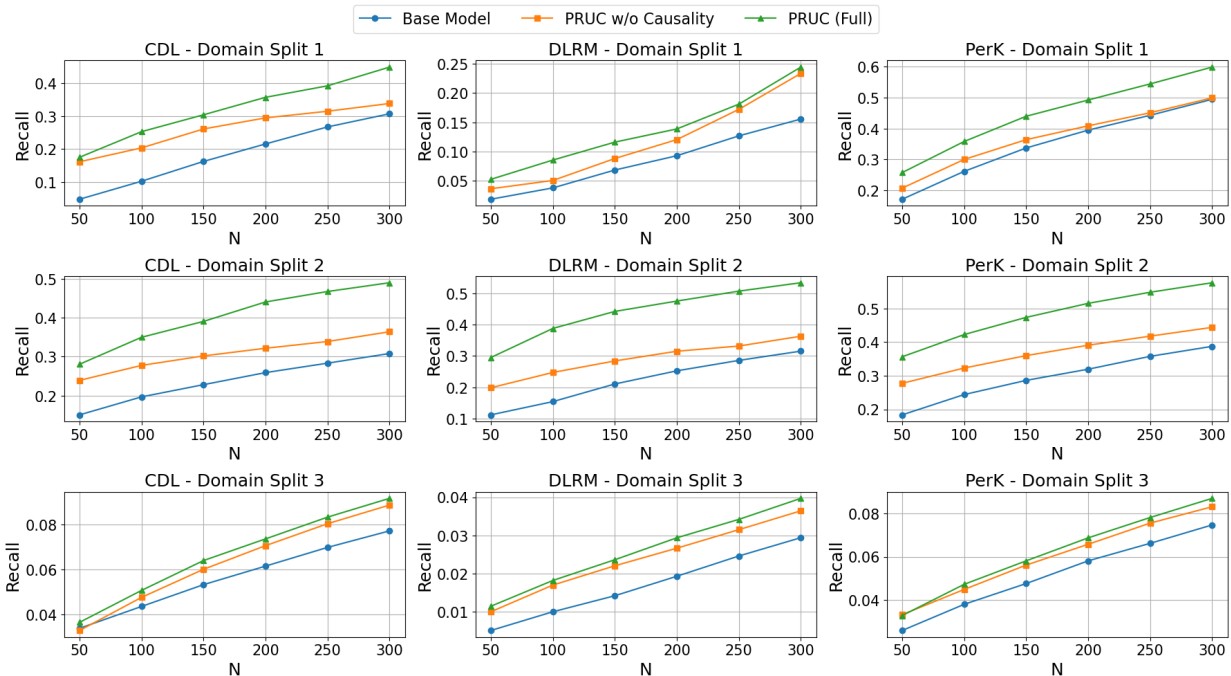

Figure 4: Recall@$N$ on XMRec for PRUC with three base models: CDL, DLRM, and PerK.

**Visualizations of the Clusters.** Fig. 5 visualizes the user latent **u** for all five base models on the XMRec dataset. Each visualization shows a distinct separation into 3 clusters, indicating successful user grouping of our model. Furthermore, Figure 6 illustrates the relationship between user clusters and items using the CDL-based PRUC model on the same dataset. For each user, we selected the item with the highest rating they have given, recorded the item ID and its rating, and visualized the results. Different clusters are represented using distinct colors, effectively showcasing the distribution and preferences of users within each cluster. For instance, Cluster 1 (Red) shows pronounced preferences for 4-5 specific items, underscoring the impact of user clustering on improving PRUC's performance.

**Performance of Each Clusters Discovered by PRUC.** For a deeper understanding of the model performance, we include more fine-grained results for different clusters discovered by PRUC in Appendix B.3. Results show that our PRUC can usually improve performance in most clusters.

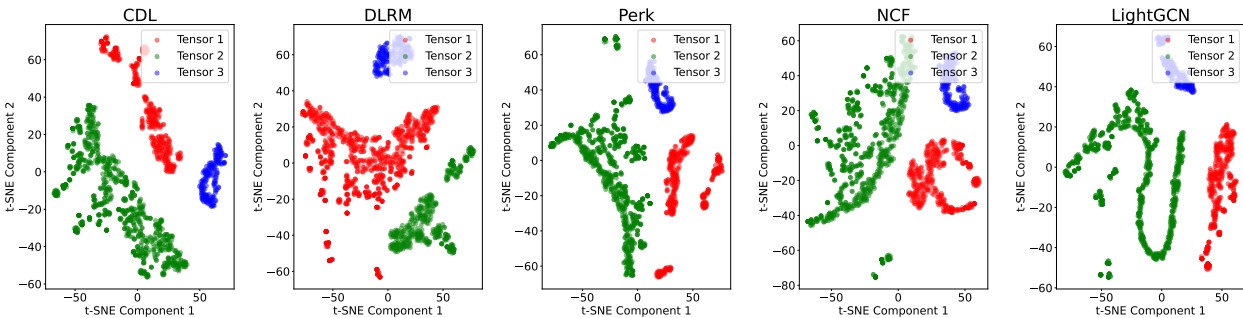

Figure 5: Clusters of users based on the user latent **u** from PRUC with base models CDL (**left**), DLRM (**left center**), PerK (**center**), NCF (**right center**) and LightGCN (**right**) for the split "France, Italy, India → Japan, Mexico". All user latents are reduced to 2D by t-SNE.

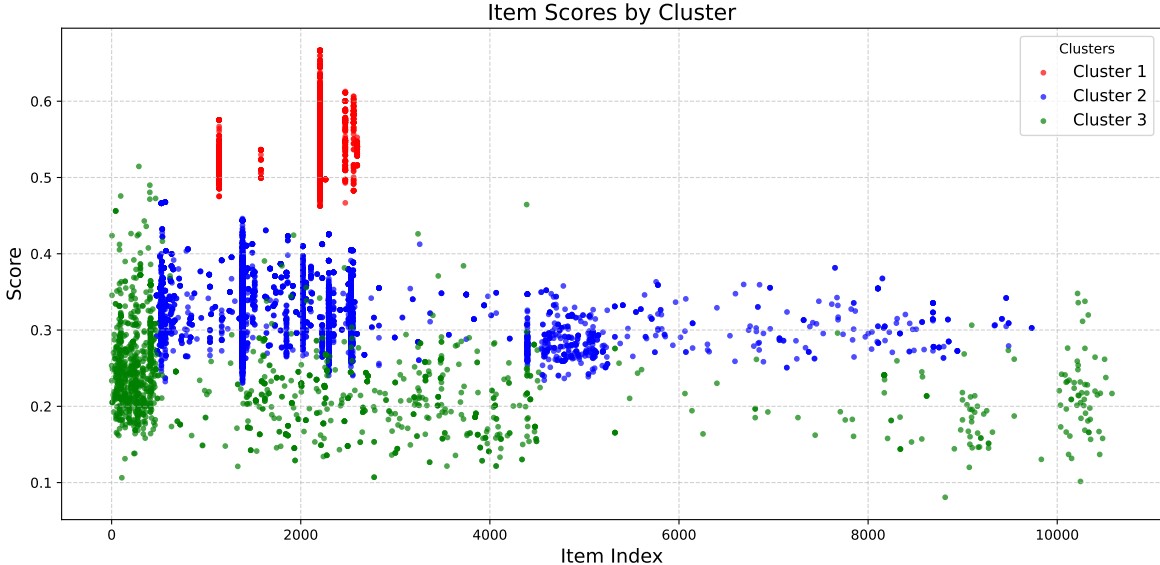

Figure 6: Clusters of users based on their highest rated items, using the CDL-based PRUC model applied to the XMRec dataset. X-axis indicates the item ID, while Y-axis indicates the score of the item. Clusters are distinguished by different colors.

**Case Study on Debiasing.** To explore PRUC's debiasing capability and cross-domain generalization, we conducted a detailed analysis using the first domain pair of the XMRec dataset, which corresponds to the countries **France, Italy, India, Japan, and Mexico**.

We first examined the top-20 recommendations for each user using the CDL baseline model. The resulting distributions showed notable country-specific biases:

- Italian users: 34 **Camera & Photo** recommendations (3.70%) out of 920 total recommendations.
- Indian users: 878 **Camera & Photo** recommendations (6.46%) out of 13,600 total recommendations.

The bias ratio across countries was $1.75\times$ (maximum: 6.46%, minimum: 3.70%), indicating that Indian users were recommended camera products 1.75 times more frequently than Italian users.

After applying our PRUC model, we examined the top-20 recommendations for each user. The resulting distributions were notably more balanced:

- Italian users: 33 **Camera & Photo** recommendations (3.59%) out of 920 total recommendations.
- Indian users: 427 **Camera & Photo** recommendations (3.14%) out of 13,600 total recommendations.

The bias ratio across countries was reduced to $1.14\times$ (maximum: $3.59\%$, minimum: $3.14\%$), representing a $38.5\%$ reduction in country-specific bias. This demonstrates that PRUC successfully mitigated the preference biases through probabilistic user clustering and causal debiasing, thereby enabling the model to learn more generalizable user-item interaction patterns.

**Ablation Study.** The comparison in Table 3, Table 4, and Fig. 4 highlights the performance difference between 'PRUC w/o Causality' and 'PRUC (Full)'. The results consistently show that 'PRUC (Full)' outperforms its counterpart, 'PRUC w/o Causality', emphasizing the crucial role of causal inference in enhancing the effectiveness of the PRUC model. Furthermore, a comparison between the base model and 'PRUC w/o Causality' also reveals notable performance improvements, validating the efficacy of PRUC's user cluster discovery. Additional details and results can be found in Table 5~18 of Appendix B.4.

## 4 Related Work

**Domain-Dependent Recommendation.** Previous work has explored various in-domain recommendation scenarios. Early methods, such as PMF (Mnih & Salakhutdinov, 2007) and BPR (Rendle et al., 2012), applied collaborative filtering techniques to address challenges in recommendation. Later, methods such as GRU4Rec (Hidasi et al., 2016), SAS4Rec (Kang & McAuley, 2018), KGAT (Wang et al., 2019), and PerK (Kweon et al., 2024) leveraged advanced deep learning models to enhance the performance of recommender systems. These approaches focus on rating data between items and users but do not incorporate item features. Collaborative deep learning (CDL) models (Wang et al., 2015; 2016; Zhang et al., 2016; Li & She, 2017) incorporate content to enable pretrained recommenders, making them more versatile in different contexts, e.g., cold start scenarios.

Despite significant advances in in-domain recommendations, cross-domain recommendation remains relatively understudied. Existing work has utilized domain adaptation techniques (Xu et al., 2023; Liu et al., 2023; Shi & Wang, 2023; Xu et al., 2022; Wang et al., 2020a; Ganin et al., 2016) to tackle this challenge, often relying on common users or items across source (training) and target (testing) domains (Yuan et al., 2020; Wu et al., 2020; Bi et al., 2020; Li et al., 2019; Hansen et al., 2020; Liang et al., 2020; Zhu et al., 2020; Liu et al., 2020). On the other hand, some methods enhance recommendation performance in both source and target domains simultaneously (Li & Tuzhilin, 2020; Hu et al., 2018; Zhao et al., 2019). In contrast to existing approaches, our PRUC first infers user clusters and confounders before making recommendations based on the identified user clusters, leading to improved generalization and robustness against domain shifts.

**Causal Inference for Recommendation.** Causal inference (Pearl, 2009) has been widely applied to model cause-and-effect relationships between variables in the machine learning community. Recently, it has been employed to improve the performance of recommender systems (Wang et al., 2020b). PDA (Zhang et al., 2021) uses causal intervention to address popularity bias in recommendations, while DICE (Zheng et al., 2021) learns representations from user interactions based on the structured causal model (SCM). Additionally, some research focuses on debiasing recommendations without adopting a causal inference perspective (Li et al., 2021; Wang et al., 2022; Chen et al., 2023). However, these approaches do not consider user groups within the SCM framework. In contrast, our method divides users into clusters based on a confounder variable and generates recommendations by aggregating user ratings through do-calculus, providing a more interpretable and sophisticated approach.

## 5 Conclusion

In this paper, we address the problem of cross-domain recommendation by introducing a novel causal Bayesian framework, named Probabilistic Residual User Clustering (PRUC). PRUC generates recommendations by: (1) inferring the user cluster ID, (2) inferring the residual rating based on our causal debiasing framework, and (3) predicting the final rating as a correction to the base model's prediction. PRUC can enhance the performance of any base recommenders in a plug-and-play manner, and automatically discover meaningful user clusters. As a general probabilistic framework compatible with various recommendation systems, PRUC can be extended to additional modalities beyond textual data in future research. Furthermore, PRUC provides interpretability by uncovering latent user preferences and biases that influence rating

predictions. Its modular design also allows seamless integration with deep learning-based recommenders, making it a scalable and adaptable solution for diverse recommendation scenarios.

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

## A  Appendix

## B  More Details on Experiments and Implementation

### B.1  Metrics

**mAP.** mAP is defined as:

$$\text{AP}_i = \frac{1}{|J_i|} \sum_{n=1}^{N} \text{rel}_{i,n} \times \text{Precision}_i @ n, \tag{31}$$

where $N$ is the total number of recommended items, $\text{Precision}_i @ n$ is the precision at rank $n$, and $|J_i|$ is the total number of relevant items for user $i$. The mean Average Precision (mAP) is then calculated by averaging $\text{AP}_i$ over all users:

$$\text{mAP} = \frac{1}{|I|} \sum_{i=1}^{|I|} \text{AP}_i, \tag{32}$$

where $|I|$ is the total number of users.

**NDCG.** NDCG@$N$ is computed as follows.

First, the Discounted Cumulative Gain (DCG@$N$) is calculated:

$$\text{DCG}_i@N = \sum_{n=1}^{N} \frac{2^{\text{rel}_{i,n}} - 1}{\log_2(n+1)}, \tag{33}$$

where $\text{rel}_{i,n}$ denotes the relevance of the item at position $n$ for user $i$. Next, the Ideal Discounted Cumulative Gain (IDCG@$N$), representing the maximum possible DCG (i.e., all relevant items ranked at the top), is calculated as:

$$\text{IDCG}_i@N = \sum_{n=1}^{\min(N,|J_i|)} \frac{2^1 - 1}{\log_2(n+1)} = \sum_{n=1}^{\min(N,|J_i|)} \frac{1}{\log_2(n+1)}, \tag{34}$$

where $|J_i|$ denotes the total number of relevant items for user $i$.

Finally, the Normalized Discounted Cumulative Gain is obtained by normalizing DCG@$N$ by IDCG@$N$:

$$\text{NDCG}_i@N = \frac{\text{DCG}_i@N}{\text{IDCG}_i@N}. \tag{35}$$

Here the logarithmic term $\log_2(n+1)$ discounts the relevance based on the item's position in the ranked list, serving as the normalization factor.

### B.2 Training Configurations

Following CDL (Wang et al., 2015), we set the hidden dimension $h = 50$ for all latent vectors, as well as for the encoder and decoder networks. During training, we use AdamW (Kingma & Ba, 2015) as our optimizer, with a learning rate of $10^{-3}$ and a batch size of 256. The base models were trained for 100 epochs, while PRUC was trained for 150 epochs. All experiments were conducted on an NVIDIA RTX A5000 GPU.

### B.3 Performance of Each Clusters Discovered by PRUC

Table 10, 11, 12, 13, 14 show PRUC's performance across different clusters on XMRec using CDL, DLRM, PerK, NCF, and LightGCN as base models. These results support the conclusion that PRUC improves upon the base models even without incorporating the causality component. Furthermore, the full PRUC consistently outperforms its non-causal counterpart across all configurations. For example, CDL, as the base model, achieves a recall@20 of 0.0241 for User Cluster 1 in the split of "France, Italy, India → Japan, Mexico". When PRUC without the causal inference component is applied, recall improves to 0.0278. The full PRUC further enhances performance for this metric, achieving a recall@20 of 0.0708.

Table 5, 6, 7, 8, 9 show PRUC's performance across different clusters on MovieLens with the same five base models. Even with some fluctuations, the similar improvements are consistent with the results for XMRec.

### B.4 Ablation Study

The performance comparison across Table 5-14 shows that "PRUC (Full)" generally outperforms "PRUC w/o Causality", highlighting the effectiveness of causal inference in PRUC. Additionally, comparing the base model with "PRUC w/o Causality" reveals performance enhancements, suggesting that PRUC's user cluster discovery significantly boosts performance.

**Simple Baseline.** Moreover, we conducted experiments by clustering the user features and then performing per-cluster modeling to predict the residuals. Table 15 presents the results comparing this simple baseline with our full PRUC, verifying that our latent variable modeling is highly effective, while per-cluster modeling alone yields limited improvements.

**Larger Base Models.** We also conducted additional ablation experiments by scaling up the CDL baseline to approximately match the parameter size of the PRUC-enriched network. Concretely, we expanded both

Table 5: Performance of PRUC on different user clusters with CDL as the base model on MovieLens. "-" means a cluster contains only training-set users, i.e., no test-set users to evaluate. The best results are marked with **bold face**.

| Data | Cluster | Method | Recall@20 | F1@20 | MAP@20 | NDCG@20 | Precision@20 |
|---|---|---|---|---|---|---|---|
| 1, 18, 35, 45, 50, 56 →25 | 1 | CDL (Base Model) | 0.0 | 0.0 | 0.0 | 0.0 | 0.0 |
| | | PRUC w/o Causality | 0.0 | 0.0 | 0.0 | 0.0 | 0.0 |
| | | PRUC (Full) | 0.0 | 0.0 | 0.0 | 0.0 | 0.0 |
| | 2 | CDL (Base Model) | - | - | - | - | - |
| | | PRUC w/o Causality | - | - | - | - | - |
| | | PRUC (Full) | - | - | - | - | - |
| | 3 | CDL (Base Model) | 0.0179 | 0.0274 | 0.0045 | 0.0581 | 0.0587 |
| | | PRUC w/o Causality | 0.0186 | 0.0302 | 0.0056 | 0.0864 | 0.0802 |
| | | PRUC (Full) | **0.0252** | **0.0409** | **0.0072** | **0.1071** | **0.1077** |
| 25 →1, 18, 35, 45, 50, 56 | 1 | CDL (Base Model) | **0.0558** | **0.0861** | **0.0174** | 0.1758 | **0.1879** |
| | | PRUC w/o Causality | 0.0317 | 0.0528 | 0.0095 | 0.1511 | 0.1572 |
| | | PRUC (Full) | **0.0558** | **0.0861** | **0.0174** | **0.1759** | **0.1879** |
| | 2 | CDL (Base Model) | 0.0651 | 0.0795 | 0.0173 | 0.0938 | 0.1020 |
| | | PRUC w/o Causality | 0.0676 | 0.0880 | 0.0183 | 0.1159 | 0.1259 |
| | | PRUC (Full) | **0.1016** | **0.1341** | **0.0319** | **0.1832** | **0.1972** |
| | 3 | CDL (Base Model) | - | - | - | - | - |
| | | PRUC w/o Causality | - | - | - | - | - |
| | | PRUC (Full) | - | - | - | - | - |

the depth and width of the CDL architecture as suggested. The original CDL structure was $512 \to 200 \to 50$, while the larger (deeper and wider) version used an architecture of $512 \to 550 \to 400 \to 50$, resulting in roughly **1.05M** parameters, which is comparable to PRUC's **0.90M** parameters. This ensures that the comparison isolates the effect of PRUC from mere model capacity differences. The results are presented in Table 16. Although the deeper CDL exhibits a slight performance improvement over the base model, PRUC still achieves substantially higher performance across most metrics. This demonstrates that PRUC's gains stem from its ability to capture cross-domain relational patterns, user-cluster-specific representations, rather than simply from an increased parameter count.

**More User Records in the Training Set.** We conducted experiments using CDL as the base model to analyze the effect of incorporating a higher number of testing-domain user records into the training process. Specifically, we used the second domain pair of the XMRec dataset, involving users from Mexico, Spain, India, Japan, and Germany. we selected users with more than four interactions and varied the number of testing-domain user records included in the training set to construct 1-shot, 2-shot, and 3-shot scenarios, where $n$-shot means there are $n$ records for each testing user in the training set. The performance of both the base CDL model and our proposed PRUC model in these settings is summarized in Table 17 and Table 18 below.

Table 6: Performance of PRUC on different user clusters with DLRM as the base model on MovieLens. "-" means a cluster contains only training-set users, i.e., no test-set users to evaluate. The best results are marked with **bold face**.

| Data | Cluster | Method | Recall@20 | F1@20 | MAP@20 | NDCG@20 | Precision@20 |
|---|---|---|---|---|---|---|---|
| 1, 18, 35, 45, 50, 56 →25 | 1 | DLRM (Base Model) - | - | - | - | - | |
| | | PRUC w/o Causality - | - | - | - | - | |
| | | PRUC (Full) - | - | - | - | - | |
| | 2 | DLRM (Base Model) | 0.0714 | 0.1097 | **0.0285** | **0.2433** | 0.2367 |
| | | PRUC w/o Causality | 0.0269 | 0.0434 | 0.0073 | 0.1078 | 0.1112 |
| | | PRUC (Full) | **0.0716** | **0.1101** | 0.0284 | 0.2431 | **0.2372** |
| | 3 | DLRM (Base Model) | 0.0 | 0.0 | 0.0 | 0.0 | 0.0 |
| | | PRUC w/o Causality | 0.0 | 0.0 | 0.0 | 0.0 | 0.0 |
| | | PRUC (Full) | 0.0 | 0.0 | 0.0 | 0.0 | 0.0 |
| 25 →1, 18, 35, 45, 50, 56 | 1 | DLRM (Base Model) | 0.0790 | 0.1264 | 0.0343 | 0.3266 | 0.3146 |
| | | PRUC w/o Causality | 0.0328 | 0.0548 | 0.0116 | 0.1716 | 0.1656 |
| | | PRUC (Full) | **0.0848** | **0.1366** | **0.0396** | **0.3634** | **0.3505** |
| | 2 | DLRM (Base Model) | 0.0882 | 0.1390 | 0.0405 | **0.3396** | **0.3271** |
| | | PRUC w/o Causality | 0.0975 | 0.1382 | 0.0426 | 0.2572 | 0.2374 |
| | | PRUC (Full) | **0.1119** | **0.1561** | **0.0486** | 0.2745 | 0.2583 |
| | 3 | DLRM (Base Model) | - | - | - | - | - |
| | | PRUC w/o Causality | - | - | - | - | - |
| | | PRUC (Full) | - | - | - | - | - |

Table 7: Performance of PRUC on different user clusters with Perk as the base model on MovieLens. "-" means a cluster contains only training-set users, i.e., no test-set users to evaluate. The best results are marked with **bold face**.

| Data | Cluster | Method | Recall@20 | F1@20 | MAP@20 | NDCG@20 | Precision@20 |
|---|---|---|---|---|---|---|---|
| 1, 18, 35, 45, 50, 56 →25 | 1 | Perk (Base Model) | - | - | - | - | - |
| | | PRUC w/o Causality | - | - | - | - | - |
| | | PRUC (Full) | - | - | - | - | - |
| | 2 | Perk (Base Model) | **0.0686** | **0.1040** | **0.0295** | **0.2271** | **0.2150** |
| | | PRUC w/o Causality | 0.0583 | 0.0884 | 0.0215 | 0.1788 | 0.1820 |
| | | PRUC (Full) | 0.0683 | 0.1036 | 0.0288 | 0.2215 | 0.2136 |
| | 3 | Perk (Base Model) | 0.0585 | 0.0745 | 0.0173 | 0.1053 | 0.1023 |
| | | PRUC w/o Causality | 0.0550 | 0.0701 | 0.0139 | 0.0942 | 0.0967 |
| | | PRUC (Full) | **0.0847** | **0.1074** | **0.0277** | **0.1559** | **0.1467** |
| 25 →1, 18, 35, 45, 50, 56 | 1 | Perk (Base Model) | **0.0745** | **0.1179** | **0.0332** | 0.2868 | 0.2826 |
| | | PRUC w/o Causality | 0.0319 | 0.0530 | 0.0140 | 0.1811 | 0.1563 |
| | | PRUC (Full) | **0.0745** | **0.1179** | **0.0332** | **0.2870** | **0.2828** |
| | 2 | Perk (Base Model) | 0.0750 | 0.1090 | 0.0292 | 0.2033 | 0.1995 |
| | | PRUC w/o Causality | 0.0939 | 0.1338 | 0.0367 | 0.2399 | 0.2323 |
| | | PRUC (Full) | **0.0984** | **0.1407** | **0.0446** | **0.2628** | **0.2469** |
| | 3 | Perk (Base Model) | - | - | - | - | - |
| | | PRUC w/o Causality | - | - | - | - | - |
| | | PRUC (Full) | - | - | - | - | - |

Table 8: Performance of PRUC on different user clusters with NCF as the base model on MovieLens. "-" means a cluster contains only training-set users, i.e., no test-set users to evaluate. The best results are marked with **bold face**.

| Data | Cluster | Method | Recall@20 | F1@20 | MAP@20 | NDCG@20 | Precision@20 |
|---|---|---|---|---|---|---|---|
| 1, 18, 35, 45, 50, 56 →25 | 1 | NCF (Base Model) | 0.0 | 0.0 | 0.0 | 0.0 | 0.0 |
| | | PRUC w/o Causality | 0.0 | 0.0 | 0.0 | 0.0 | 0.0 |
| | | PRUC (Full) | **0.1964** | **0.1325** | **0.0230** | **0.0754** | **0.1000** |
| | 2 | NCF (Base Model) | 0.0051 | 0.0087 | 0.0011 | 0.0282 | 0.0279 |
| | | PRUC w/o Causality | 0.0271 | 0.0443 | 0.0067 | 0.1134 | 0.1210 |
| | | PRUC (Full) | **0.0285** | **0.0463** | **0.0070** | **0.1159** | **0.1231** |
| | 3 | NCF (Base Model) | 0.0047 | 0.0074 | 0.0009 | 0.0172 | 0.0177 |
| | | PRUC w/o Causality | **0.0125** | **0.0192** | **0.0023** | 0.0386 | 0.0409 |
| | | PRUC (Full) | 0.0120 | 0.0185 | 0.0022 | **0.0389** | **0.0410** |
| 25 →1, 18, 35, 45, 50, 56 | 1 | NCF (Base Model) | 0.0149 | 0.0248 | 0.0032 | 0.0710 | 0.0729 |
| | | PRUC w/o Causality | **0.0309** | **0.0515** | **0.0088** | **0.1494** | **0.1555** |
| | | PRUC (Full) | 0.0306 | 0.0512 | 0.0087 | 0.1484 | 0.1551 |
| | 2 | NCF (Base Model) | - | - | - | - | - |
| | | PRUC w/o Causality | - | - | - | - | - |
| | | PRUC (Full) | - | - | - | - | - |
| | 3 | NCF (Base Model) | 0.0098 | 0.0150 | 0.0021 | 0.0302 | 0.0319 |
| | | PRUC w/o Causality | 0.0941 | 0.1316 | 0.0312 | 0.2094 | 0.2185 |
| | | PRUC (Full) | **0.1071** | **0.1481** | **0.0392** | **0.2309** | **0.2402** |

Table 9: Performance of PRUC on different user clusters with LightGCN as the base model on MovieLens. "-" means a cluster contains only training-set users, i.e., no test-set users to evaluate. The best results are marked with **bold face**.

| Data | Cluster | Method | Recall@20 | F1@20 | MAP@20 | NDCG@20 | Precision@20 |
|---|---|---|---|---|---|---|---|
| 1, 18, 35, 45, 50, 56 →25 | 1 | LightGCN (Base Model) | - | - | - | - | - |
| | | PRUC w/o Causality | - | - | - | - | - |
| | | PRUC (Full) | - | - | - | - | - |
| | 2 | LightGCN (Base Model) | 0.0081 | 0.0132 | 0.0019 | 0.0381 | 0.0358 |
| | | PRUC w/o Causality | **0.0248** | **0.0402** | **0.0070** | **0.1075** | 0.1052 |
| | | PRUC (Full) | **0.0248** | 0.0401 | 0.0069 | 0.1073 | **0.1053** |
| | 3 | LightGCN (Base Model) | 0.0226 | 0.0224 | 0.0075 | 0.0227 | 0.0222 |
| | | PRUC w/o Causality | 0.0214 | 0.0378 | 0.0115 | 0.1911 | 0.1611 |
| | | PRUC (Full) | **0.0563** | **0.0884** | **0.0226** | **0.2219** | **0.2056** |
| 25 →1, 18, 35, 45, 50, 56 | 1 | LightGCN (Base Model) | 0.0094 | 0.0157 | 0.0022 | 0.0498 | 0.0484 |
| | | PRUC w/o Causality | **0.0300** | **0.0495** | **0.0101** | **0.1515** | **0.1416** |
| | | PRUC (Full) | 0.0288 | 0.0477 | 0.0097 | 0.1492 | 0.1394 |
| | 2 | LightGCN (Base Model) | 0.0068 | 0.0110 | 0.0011 | 0.0277 | 0.0294 |
| | | PRUC w/o Causality | 0.0297 | 0.0428 | 0.0130 | 0.0953 | 0.0765 |
| | | PRUC (Full) | **0.0531** | **0.0793** | **0.0204** | **0.1597** | **0.1559** |
| | 3 | LightGCN (Base Model) | - | - | - | - | - |
| | | PRUC w/o Causality | - | - | - | - | - |
| | | PRUC (Full) | - | - | - | - | - |

Table 10: Performance of PRUC on different user clusters with CDL as the base model on XMRec. "-" means a cluster contains only training-set users, i.e., no test-set users to evaluate. The best results are marked with **bold face**.

| Data | Cluster | Method | Recall@20 | F1@20 | MAP@20 | NDCG@20 | Precision@20 |
|---|---|---|---|---|---|---|---|
| | 1 | CDL (Base Model) | 0.0241 | 0.0028 | 0.0062 | 0.0018 | 0.0015 |
| | | PRUC w/o Causality | **0.1972** | **0.0238** | **0.0905** | **0.0197** | **0.0127** |
| | | PRUC (Full) | 0.0708 | 0.0074 | 0.0652 | 0.0105 | 0.0039 |
| France, Italy, India →Japan, Mexico | 2 | CDL (Base Model) | 0.0126 | 0.0014 | 0.0022 | 0.0007 | 0.0008 |
| | | PRUC w/o Causality | 0.0902 | 0.0107 | 0.0236 | 0.0069 | 0.0057 |
| | | PRUC (Full) | **0.1156** | **0.0138** | **0.0431** | **0.0109** | **0.0073** |
| | 3 | CDL (Base Model) | - | - | - | - | - |
| | | PRUC w/o Causality | - | - | - | - | - |
| | | PRUC (Full) | - | - | - | - | - |
| | 1 | CDL (Base Model) | 0.1742 | 0.0225 | 0.0333 | 0.0123 | 0.0120 |
| | | PRUC w/o Causality | **0.2114** | **0.0267** | **0.0707** | **0.0194** | **0.0142** |
| | | PRUC (Full) | 0.1665 | 0.0222 | 0.0634 | 0.0170 | 0.0119 |
| Mexico, Spain, India →Japan, Germany | 2 | CDL (Base Model) | 0.0903 | 0.0102 | 0.0289 | 0.0072 | 0.0054 |
| | | PRUC w/o Causality | 0.1532 | 0.0187 | 0.0524 | 0.0136 | 0.0100 |
| | | PRUC (Full) | **0.1796** | **0.0233** | **0.0579** | **0.0160** | **0.0124** |
| | 3 | CDL (Base Model) | - | - | - | - | - |
| | | PRUC w/o Causality | - | - | - | - | - |
| | | PRUC (Full) | - | - | - | - | - |
| | 1 | CDL (Base Model) | 0.0262 | 0.0059 | **0.0079** | 0.0041 | 0.0033 |
| | | PRUC w/o Causality | 0.0261 | 0.0063 | 0.0072 | **0.0044** | 0.0036 |
| | | PRUC (Full) | **0.0266** | **0.0064** | 0.0062 | 0.0042 | **0.0037** |
| Germany, Italy, Japan →United States, India | 2 | CDL (Base Model) | 0.0244 | 0.0054 | **0.0088** | 0.0042 | **0.0031** |
| | | PRUC w/o Causality | 0.0166 | 0.0037 | 0.0041 | 0.00234 | 0.0021 |
| | | PRUC (Full) | **0.0250** | **0.0055** | **0.0088** | **0.0042** | **0.0031** |
| | 3 | CDL (Base Model) | 0.0277 | **0.0049** | 0.0066 | 0.0028 | **0.0027** |
| | | PRUC w/o Causality | 0.0194 | 0.0045 | 0.0049 | **0.0030** | 0.0026 |
| | | PRUC (Full) | **0.0278** | **0.0049** | **0.0067** | 0.0028 | **0.0027** |

Table 11: Performance of PRUC on different user clusters with DLRM as the base model on XMRec. "-" means a cluster contains only training-set users, i.e., no test-set users to evaluate. The best results are marked with **bold face**.

| Data | Cluster | Method | Recall@20 | F1@20 | MAP@20 | NDCG@20 | Precision@20 |
|---|---|---|---|---|---|---|---|
| | 1 | DLRM (Base Model) | 0.0051 | 0.0005 | 0.0004 | 0.0002 | 0.0003 |
| | | PRUC w/o Causality | 0.0246 | 0.0027 | 0.0039 | 0.0014 | 0.0014 |
| | | PRUC (Full) | **0.0345** | **0.004** | **0.0056** | **0.0021** | **0.0021** |
| France, Italy, India →Japan, Mexico | 2 | DLRM (Base Model) | 0.0000 | 0.0000 | 0.0000 | 0.0000 | 0.0000 |
| | | PRUC w/o Causality | **0.0150** | **0.0017** | **0.0040** | **0.0010** | **0.0009** |
| | | PRUC (Full) | 0.0000 | 0.0000 | 0.0000 | 0.0000 | 0.0000 |
| | 3 | DLRM (Base Model) | - | - | - | - | - |
| | | PRUC w/o Causality | - | - | - | - | - |
| | | PRUC (Full) | - | - | - | - | - |
| | 1 | DLRM (Base Model) | 0.0000 | 0.0000 | 0.0000 | 0.0000 | 0.0000 |
| | | PRUC w/o Causality | **0.3296** | **0.0416** | 0.0203 | **0.0153** | **0.0222** |
| | | PRUC (Full) | 0.3074 | 0.0395 | **0.0213** | 0.0152 | 0.0211 |
| Mexico, Spain, India →Japan, Germany | 2 | DLRM (Base Model) | 0.0780 | 0.0096 | 0.0087 | 0.0042 | 0.0051 |
| | | PRUC w/o Causality | 0.1398 | 0.0174 | 0.0277 | 0.0096 | 0.0093 |
| | | PRUC (Full) | **0.1984** | **0.0241** | **0.0555** | **0.0157** | **0.0128** |
| | 3 | DLRM (Base Model) | - | - | - | - | - |
| | | PRUC w/o Causality | - | - | - | - | - |
| | | PRUC (Full) | - | - | - | - | - |
| | 1 | DLRM (Base Model) | 0.0023 | 0.0006 | 0.0003 | 0.0003 | 0.0003 |
| | | PRUC w/o Causality | 0.0042 | **0.0011** | **0.0010** | **0.0007** | **0.0007** |
| | | PRUC (Full) | **0.0046** | **0.0011** | 0.0009 | **0.0007** | 0.0006 |
| Germany, Italy, Japan →United States, India | 2 | DLRM (Base Model) | 0.0018 | 0.0005 | 0.0003 | 0.0003 | 0.0003 |
| | | PRUC w/o Causality | **0.0045** | **0.0012** | 0.0010 | **0.0007** | **0.0007** |
| | | PRUC (Full) | **0.0045** | 0.0011 | **0.0012** | **0.0007** | **0.0007** |
| | 3 | DLRM (Base Model) | 0.0036 | 0.0008 | 0.0005 | 0.0004 | 0.0004 |
| | | PRUC w/o Causality | 0.0052 | 0.0015 | 0.0009 | 0.0009 | 0.0009 |
| | | PRUC (Full) | **0.0141** | **0.0034** | **0.0075** | **0.0032** | **0.0019** |

Table 12: Performance of PRUC on different user clusters with PerK as the base model on XMRec. "-" means a cluster contains only training-set users, i.e., no test-set users to evaluate. The best results are marked with **bold face**.

| Data | Cluster | Method | Recall@20 | F1@20 | MAP@20 | NDCG@20 | Precision@20 |
|---|---|---|---|---|---|---|---|
| France, Italy, India →Japan, Mexico | 1 | PerK (Base Model)
PRUC w/o Causality
PRUC (Full) | 0.1752
**0.2153**
0.1782 | 0.0204
**0.0260**
0.0210 | 0.1152
**0.1255**
0.1162 | 0.022
**0.0252**
0.0226 | 0.0108
**0.0139**
0.0114 |
| | 2 | PerK (Base Model)
PRUC w/o Causality
PRUC (Full) | 0.0986
0.1243
**0.1629** | 0.0115
0.0143
**0.0189** | 0.0403
0.0440
**0.0548** | 0.0094
0.0108
**0.0138** | 0.0061
0.0076
**0.0100** |
| | 3 | PerK (Base Model)
PRUC w/o Causality
PRUC (Full) | -
-
- | -
-
- | -
-
- | -
-
- | -
-
- |
| Mexico, Spain, India →Japan, Germany | 1 | PerK (Base Model)
PRUC w/o Causality
PRUC (Full) | 0.1434
0.2175
**0.2905** | 0.0176
0.0262
**0.0353** | 0.0582
0.0913
**0.1157** | 0.014
0.0217
**0.0278** | 0.0094
0.0140
**0.0188** |
| | 2 | PerK (Base Model)
PRUC w/o Causality
PRUC (Full) | 0.1495
**0.2783**
0.1790 | 0.0184
**0.0307**
0.0224 | 0.0723
**0.0964**
0.0646 | 0.0166
**0.0232**
0.0167 | 0.0098
**0.0163**
0.0120 |
| | 3 | PerK (Base Model)
PRUC w/o Causality
PRUC (Full) | -
-
- | -
-
- | -
-
- | -
-
- | -
-
- |
| Germany, Italy, Japan →United States, India | 1 | PerK (Base Model)
PRUC w/o Causality
PRUC (Full) | 0.0194
0.0295
**0.0308** | 0.0043
0.0066
**0.0068** | 0.0057
**0.0087**
0.0086 | 0.003
**0.0046**
**0.0046** | 0.0024
0.0037
**0.0038** |
| | 2 | PerK (Base Model)
PRUC w/o Causality
PRUC (Full) | 0.0126
0.0155
**0.0162** | 0.0028
0.0035
**0.0037** | 0.0032
0.0040
**0.0048** | 0.0018
0.0022
**0.0025** | 0.0016
0.0020
**0.0021** |
| | 3 | PerK (Base Model)
PRUC w/o Causality
PRUC (Full) | 0.0261
0.0174
**0.0266** | 0.0035
0.0027
**0.0041** | 0.0091
0.0013
**0.0102** | 0.0025
0.0012
**0.0033** | 0.0019
0.0014
**0.0022** |

Table 13: Performance of PRUC on different user clusters with NCF as the base model on XMRec. "-" means a cluster contains only training-set users, i.e., no test-set users to evaluate. The best results are marked with **bold face**.

| Data | Cluster | Method | Recall@20 | F1@20 | MAP@20 | NDCG@20 | Precision@20 |
|---|---|---|---|---|---|---|---|
| France, Italy, India →Japan, Mexico | 1 | NCF (Base Model)
PRUC w/o Causality
PRUC (Full) | 0.0090
**0.2013**
0.1581 | 0.0010
**0.0238**
0.0176 | 0.0019
**0.0537**
0.0476 | 0.0005
**0.0151**
0.0122 | 0.0005
**0.0127**
0.0093 |
| | 2 | NCF (Base Model)
PRUC w/o Causality
PRUC (Full) | 0.0165
0.0893
**0.1062** | 0.0019
0.0107
**0.0130** | 0.0032
0.0184
**0.0280** | 0.0010
0.0061
**0.0084** | 0.0010
0.0057
**0.0069** |
| | 3 | NCF (Base Model)
PRUC w/o Causality
PRUC (Full) | -
-
- | -
-
- | -
-
- | -
-
- | -
-
- |
| Mexico, Spain, India →Japan, Germany | 1 | NCF (Base Model)
PRUC w/o Causality
PRUC (Full) | 0.0097
0.1081
**0.1560** | 0.0013
0.0142
**0.0202** | 0.0022
0.0181
**0.0280** | 0.0007
0.0073
**0.0107** | 0.0007
0.0076
**0.0108** |
| | 2 | NCF (Base Model)
PRUC w/o Causality
PRUC (Full) | -
-
- | -
-
- | -
-
- | -
-
- | -
-
- |
| | 3 | NCF (Base Model)
PRUC w/o Causality
PRUC (Full) | 0.0
0.0
0.0 | 0.0
0.0
0.0 | 0.0
0.0
0.0 | 0.0
0.0
0.0 | 0.0
0.0
0.0 |
| Germany, Italy, Japan →United States, India | 1 | NCF (Base Model)
PRUC w/o Causality
PRUC (Full) | 0.0020
0.0204
**0.0214** | 0.0005
0.0051
**0.0055** | 0.0006
**0.0041**
0.0039 | 0.0004
0.0030
**0.0032** | 0.0003
0.0029
**0.0031** |
| | 2 | NCF (Base Model)
PRUC w/o Causality
PRUC (Full) | 0.0018
0.0064
**0.0079** | 0.0005
0.0015
**0.0021** | 0.0003
0.0008
**0.0011** | 0.0003
0.0008
**0.0011** | 0.0003
0.0009
**0.0012** |
| | 3 | NCF (Base Model)
PRUC w/o Causality
PRUC (Full) | 0.0
0.0
0.0 | 0.0
0.0
0.0 | 0.0
0.0
0.0 | 0.0
0.0
0.0 | 0.0
0.0
0.0 |

Table 14: Performance of PRUC on different user clusters with LightGCN as the base model on XMRec. "-" means a cluster contains only training-set users, i.e., no test-set users to evaluate. The best results are marked with **bold face**.

| Data | Cluster | Method | Recall@20 | F1@20 | MAP@20 | NDCG@20 | Precision@20 |
|---|---|---|---|---|---|---|---|
| France, Italy, India →Japan, Mexico | 1 | LightGCN (Base Model) | 0.0261 | 0.0034 | 0.0028 | 0.0015 | 0.0018 |
| | | PRUC w/o Causality | **0.1742** | **0.0209** | **0.0749** | **0.0167** | **0.0111** |
| | | PRUC (Full) | 0.1400 | 0.0154 | 0.0482 | 0.0115 | 0.0081 |
| | 2 | LightGCN (Base Model) | 0.0168 | 0.0019 | 0.0054 | 0.0013 | 0.0010 |
| | | PRUC w/o Causality | 0.0804 | 0.0095 | 0.0211 | 0.0060 | 0.0051 |
| | | PRUC (Full) | **0.0936** | **0.0115** | **0.0288** | **0.0079** | **0.0061** |
| | 3 | LightGCN (Base Model) | - | - | - | - | - |
| | | PRUC w/o Causality | - | - | - | - | - |
| | | PRUC (Full) | - | - | - | - | - |
| Mexico, Spain, India →Japan, Germany | 1 | LightGCN (Base Model) | 0.0093 | 0.0009 | 0.0046 | 0.0008 | 0.0005 |
| | | PRUC w/o Causality | **0.0972** | **0.0097** | **0.0129** | **0.0045** | **0.0051** |
| | | PRUC (Full) | 0.0 | 0.0 | 0.0 | 0.0 | 0.0 |
| | 2 | LightGCN (Base Model) | 0.0170 | 0.0023 | 0.0062 | 0.0017 | 0.0012 |
| | | PRUC w/o Causality | 0.1040 | 0.0135 | 0.0215 | 0.0077 | 0.0072 |
| | | PRUC (Full) | **0.1790** | **0.0224** | **0.0646** | **0.0167** | **0.0120** |
| | 3 | LightGCN (Base Model) | - | - | - | - | - |
| | | PRUC w/o Causality | - | - | - | - | - |
| | | PRUC (Full) | - | - | - | - | - |
| Germany, Italy, Japan →United States, India | 1 | LightGCN (Base Model) | 0.0016 | 0.0005 | 0.0002 | 0.0002 | 0.0003 |
| | | PRUC w/o Causality | 0.0062 | **0.0017** | 0.0012 | 0.0010 | **0.0010** |
| | | PRUC (Full) | **0.0066** | **0.0017** | 0.0014 | **0.0011** | 0.0010 |
| | 2 | LightGCN (Base Model) | 0.0 | 0.0 | 0.0 | 0.0 | 0.0 |
| | | PRUC w/o Causality | 0.0 | 0.0 | 0.0 | 0.0 | 0.0 |
| | | PRUC (Full) | 0.0 | 0.0 | 0.0 | 0.0 | 0.0 |
| | 3 | LightGCN (Base Model) | 0.0016 | 0.0004 | .0002 | 0.0002 | 0.0002 |
| | | PRUC w/o Causality | 0.0037 | **0.0009** | 0.0010 | **0.0006** | **0.0005** |
| | | PRUC (Full) | **0.0039** | 0.0008 | **0.0012** | **0.0006** | **0.0005** |

Table 15: Results on the simple baseline and PRUC.

| Method | Recall@20 | F1@20 | MAP@20 | NDCG@20 | Precision@20 |
|---|---|---|---|---|---|
| CDL | 0.0143 | 0.0016 | 0.0028 | 0.0009 | 0.0009 |
| Clustering + Residuals | 0.0156 | 0.0018 | 0.0023 | 0.0009 | 0.0010 |
| PRUC (Full) | **0.1091** | **0.0128** | **0.0463** | **0.0108** | **0.0068** |
| DLRM | 0.0044 | 0.0004 | 0.0004 | 0.0002 | 0.0002 |
| Clustering + Residuals | 0.0163 | 0.0018 | 0.0029 | 0.0009 | 0.0010 |
| PRUC (Full) | **0.0295** | **0.0035** | **0.0048** | **0.0018** | **0.0018** |
| PerK | 0.1098 | 0.0128 | 0.0512 | 0.0112 | 0.0068 |
| Clustering + Residuals | 0.1118 | 0.0129 | 0.0513 | 0.0113 | 0.0069 |
| PRUC (Full) | **0.1635** | **0.0192** | **0.0637** | **0.0151** | **0.0102** |
| NCF | 0.0131 | 0.00148 | 0.0026 | 0.0008 | 0.0008 |
| Clustering + Residuals | 0.0164 | 0.0019 | 0.0029 | 0.0010 | 0.0010 |
| PRUC (Full) | **0.1137** | **0.0137** | **0.0309** | **0.0090** | **0.0073** |
| LightGCN | 0.0182 | 0.0021 | 0.0050 | 0.0014 | 0.0011 |
| Clustering + Residuals | 0.0277 | 0.0031 | 0.0054 | 0.0017 | 0.0017 |
| PRUC (Full) | **0.1003** | **0.0121** | **0.0316** | **0.0084** | **0.0064** |

Table 16: Performance comparison between larger CDL and PRUC.

| Method | Recall@20 | F1@20 | MAP@20 | NDCG@20 | Precision@20 |
|---|---|---|---|---|---|
| CDL | 0.0143 | 0.0016 | 0.0028 | 0.0009 | 0.0009 |
| CDL (Larger) | 0.0223 | 0.0026 | 0.0022 | 0.0011 | 0.0014 |
| PRUC (Full) | **0.1091** | **0.0128** | **0.0463** | **0.0108** | **0.0068** |

Table 17: Results in $n$-shot settings for the CDL base model.

| Method | Config | Recall@20 | Precision@20 | F1@20 | MAP@20 | NDCG@20 |
|---|---|---|---|---|---|---|
| CDL (Base Model) | 1-shot | 0.0647 | **0.0105** | 0.0181 | 0.0111 | 0.0098 |
| CDL (Base Model) | 2-shot | 0.0700 | 0.0085 | **0.0152** | 0.0108 | 0.0078 |
| CDL (Base Model) | 3-shot | **0.0817** | 0.0060 | 0.0112 | **0.0159** | **0.0106** |

Table 18: Results in $n$-shot settings for our proposed model PRUC (Full) using CDL as the base model.

| Method | Config | Recall@20 | Precision@20 | F1@20 | MAP@20 | NDCG@20 |
|---|---|---|---|---|---|---|
| PRUC (Full) | 1-shot | 0.1080 | **0.0195** | **0.0359** | 0.0252 | 0.0208 |
| PRUC (Full) | 2-shot | 0.1507 | **0.0195** | 0.0356 | 0.0359 | 0.0216 |
| PRUC (Full) | 3-shot | **0.2178** | 0.0145 | 0.0272 | **0.0471** | **0.0251** |

