# OpenReview forum: "Probabilistic Residual User Clustering"
_TMLR — Rejected by TMLR_

### Review · Reviewer_nFkS · 2025-09-26

**Summary Of Contributions:**

1. The paper proposes PRUC, a causal hierarchical Bayesian, plug-and-play framework that augments any base recommender by (i) predicting residual ratings, (ii) discovering latent user clusters with a Gaussian mixture, and (iii) debiasing via do-calculus over latent confounders in a structured causal model (SCM).
2. Methodologically, PRUC performs causal Bayesian model averaging to estimate the residual while intervening on user/item representations to cut confounder influence.
3. The paper evaluates PRUC on XMRec and MovieLens under a cold-start cross-domain setup, reporting consistent gains across five base recommenders.

**Additional Comments:**

I have no additional comments.

**Audience:**

Yes

**Audience Explanation:**

The work sits at the intersection of recommender systems, probabilistic modeling, and causal inference, offering a general, base-agnostic residual wrapper that improves and interprets existing models. It is a combination of ideas of clear interest to TMLR’s readership.

**Broader Impact Concerns:**

I have no concerns.

**Claims And Evidence:**

Yes

**Claims Explanation:**

1. Table 1 shows sizable improvements over CDL and DLRM when wrapped by PRUC. On MovieLens, PRUC improves DLRM, PerK, NCF, and LightGCN.
2. Ablations indicate PRUC (Full) > PRUC w/o Causality across settings, supporting the causal component’s contribution.

**Requested Changes:**

1. I believe this work specifically focuses on cross-domain recommendation. This can be more clarified in Section 1 and 2.1.
2. This work well describes the technical details of the proposed approach, but a more important part is missing; Why do we model the probabilistic model as in Figure 2, and choose the parametric distributions as described? Aren’t there any alternatives? In what sense is the proposed approach better than existing similar approaches? The design choices of the proposed approach should be better justified.
3. This field is evolving fast, and there are numerous recent models for recommendation. It is unclear how the authors chose these competitors, which are not new.
4. The two included datasets include many different domains. Among all possible domain transfer scenarios, it is unclear why the authors chose the specific ones.
5. Please report training/inference runtime and memory costs of adding PRUC (K sub-models + EM-style training), and compare to base models alone.

---

> ### Author Response · Authors · 2025-11-14
> **Thank You for Your Encouraging and Construcive Comments [1/2]**
>
> Thank you for your encouraging comments. We are glad that you found our idea ``"of clear interest to TMLR’s readership"`` and that our experiments showing ``"consistent gains across five base recommenders"`` and ``"sizable improvements over CDL and DLRM"``. Below, we address your questions one by one. We have revised the manuscript accordingly, with all changes marked in blue.
>
> ## Requested Changes
> **Q1: "I believe this work specifically focuses on cross-domain recommendation. This should be clarified more explicitly in Section 1 and Section 2.1.""**
>
> This is a good suggestion. We have made this clearer in Section 1 and Section 2.1 as suggested.
>
> **Q2: "This work well describes the technical details of the proposed approach, but a more important part is missing; Why do we model the probabilistic model as in Figure 2, and choose the parametric distributions as described? Aren’t there any alternatives? In what sense is the proposed approach better than existing similar approaches? The design choices of the proposed approach should be better justified."**
>
> Thank you for mentioning this. To the best of our knowledge, our PRUC is the first plug-and-play method that clusters users to predict the residual rating and thereby improve the base recommender's performance.
>
> **The Need for Probabilistic Modeling in Figure 2.** Following your suggestion (and inspired by Reviewer c93F), to further justify the need for the probabilistic models in Figure 2 we ran additional experiments with a simple baseline that (1) directly clusters users without any probabilistic models to predict the residual ratings and then (2) combines these residual ratings with the base recommender's predictions to produce the final predicted ratings.
>
> Our preliminary experiments showed that such a simple baseline cannot achieve satisfactory performance. For example, Table A below shows some results comparing this **simple baseline, i.e., "Clustering + Residuals"** (as a natural "alternative" as you mentioned) with our full PRUC, verifying such latent variable probabilistic modeling is necessary and well justified.
>
> We have incorporated these new results in the revision in case future readers are interested.
>
> Table A. Results on the simple baseline and PRUC.
> | Method | Recall@20 | F1@20 | MAP@20 | NDCG@20 | Precision@20 |
> |---------|------------|--------|----------|-------------|---------------|
> | CDL (Base Model) | 0.0143 | 0.0016 | 0.0028 | 0.0009 | 0.0009 |
> | CDL (Clustering + Residuals) | 0.0156 | 0.0018 | 0.0023 | 0.0009 | 0.0010 |
> | PRUC (Full) | **0.1091** | **0.0128** | **0.0463** | **0.0108** | **0.0068** |
> ||
> | DLRM (Base Model) | 0.0044 | 0.0004 | 0.0004 | 0.0002 | 0.0002 |
> | DLRM (Clustering + Residuals) | 0.0163 | 0.0018 | 0.0029 | 0.0009 | 0.0010 |
> | PRUC (Full) | **0.0295** | **0.0035** | **0.0048** | **0.0018** | **0.0018** |
> ||
> | PerK (Base Model) | 0.1098 | 0.0128 | 0.0512 | 0.0112 | 0.0068 |
> | PerK (Clustering + Residuals) | 0.1118 | 0.0129 | 0.0513 | 0.0113 | 0.0069 |
> | PRUC (Full) | **0.1635** | **0.0192** | **0.0637** | **0.0151** | **0.0102** |
> ||
> | NCF (Base Model) | 0.0131 | 0.00148 | 0.0026 | 0.0008 | 0.0008 |
> | NCF (Clustering + Residuals) | 0.0164 | 0.0019 | 0.0029 | 0.0010 | 0.0010 |
> | PRUC (Full) | **0.1137** | **0.0137** | **0.0309** | **0.0090** | **0.0073** |
> ||
> | LightGCN (Base Model) | 0.0182 | 0.0021 | 0.0050 | 0.0014 | 0.0011 |
> | LightGCN (Clustering + Residuals) | 0.0277 | 0.0031 | 0.0054 | 0.0017 | 0.0017 |
> | PRUC (Full) | **0.1003** | **0.0121** | **0.0316** | **0.0084** | **0.0064** |
>
> **The Need for Causal Modeling in Figure 2.** As you mentioned, our ablation studies in Table 3 and Table 4 of the paper ``"indicate PRUC (Full) > PRUC w/o Causality across settings, supporting the causal component’s contribution"`` and therefore verify the need for our causal modeling.
>
>
> **Q3: "The field is evolving fast, and there are numerous recent models for recommendation. It is unclear how the authors chose the competitors, which are not new."**
>
>
> This is a good question. We would like to clarify that our baselines do cover both state-of-the-art methods and classic methods that are commonly used and well recognized. For example, among our baselines (and base models):
>
> + PerK is a state-of-the-art recommender system published at WWW 2024.
>
> + CDL (KDD 2015) is the first hybrid deep recommender system; it is a commonly used baseline that is widely recognized in the field of recommender systems. It has over 2,000 citations, partially deployed at Amazon Personalize, and was recognized as the KDD 2025 Test of Time Award.
>
> + NCF (WWW 2017) is another commonly used deep recommender system baseline, with over 8,000 citations to date.
>
> + DLRM is a production-grade recommender system, deployed in Meta.
>
> + LightGCN (SIGIR 2020) is one of the most commonly used baselines that are based on graph convolutional networks (GCN).

---

> ### Author Response · Authors · 2025-11-14
> **Thank You for Your Encouraging and Construcive Comments [2/2]**
>
> **Q4: "The two included datasets cover many different domains. Among all possible domain transfer scenarios, it is unclear why the authors chose the specific ones.**
>
> We chose MovieLens because
> + it is the most commonly used dataset for recommender systems and
> + it happens to contain "movie year" for each movie and therefore naturally forms a dataset with multiple domains, where movies from the same year belong to the same domain.
>
> We chose XMRec because it is the largest public recommendation dataset that spans 18 countries, 16 product categories, and over 52 million user-item interactions. Data from 18 different countries naturally form 18 domains, making it another perfect dataset for our experiments on cross-domain recommendation.
>
>
> **Q5: "Please report training/inference runtime and memory costs of adding PRUC (K sub-models + EM-style training), and compare these to the base models alone."**
>
>
> Thank you for mentioning this. Following your suggestion, we have run additional experiments to evaluate the training and inference time as well as memory cost of adding PRUC.
>
> Specifically, we conducted the experiment using the first domain pair of the XMRec dataset (in Table 1), which corresponds to the countries France, Italy, India, Japan, and Mexico. The dataset contains a total number of 1567 users and 1324 items, with interactions used for training the CDL model. We compare the training time, inference time and GPU Peak Memory between CDL and PRUC training on CDL. Table B below summarizes the results.
>
> Table B. Time and memory costs of adding PRUC
>
> | Metric                  | Baseline | PRUC (K=3) |
> |--------------------------|-----------|-------------|
> | Training Time (s)        | 116.64    | 496.44      |
> | Inference Time (s)       | 0.97      | 4.62        |
> | GPU Peak Memory (MB)     | 13.8      | 22.4        |
>
> Last but not least, we would like to thank Reviewer nFkS for the careful review and constructive comments, especially on the clarification of existing baselines and suggestion of an additional simple baseline; they have further highlighted the effectiveness of our method and the necessity of probabilistic modeling.
>
> We believe all concerns have now been addressed. Please do not hesitate to let us know if you have any follow-up questions, which we will be more than happy to address.

---

### Review · Reviewer_c93F · 2025-10-24

**Summary Of Contributions:**

This paper looks at the problem of applying recommender systems across multiple domains. This is difficult when such systems are "cold started" or transfered to new markets with no users/items in common. The proposed approach is a hierarchical probabilistic model that explains user embeddings in terms of K latent clusters shared across all domains. The per-user embeddings are impacted by a latent causal confounder s_m that is specific to each domain m. The model in turn generates user embeddings, item embeddings, ratings, and observed per-item features (see PGM on Eq 3).

Using this assumed model, they describe a procedure to estimate the posterior over user-embeddings u_i and item-embeddings v_j given observed per-item features x_j and ratings R_ij. They use pretty standard variational methods (in tradition of Blei et al. JASA 2017), with a factorized posterior $q(u)q(v)$.

They then use the estimated cluster assignments to predict residual ratings: each cluster has a specific prediction model that maps from u_i, v_j, and confounders s to the residual rating (see eq 25).

Experiments examine two large recommendation tasks, where the domains are age groups for the movie reviews and countries for a product purchase tasks.

### Overall Strengths

* I like the modeling approach overall... the idea that latent clusters can impact residuals is interesting and seems to have potential in this application.
* Experiments seem to be looking at reasonable datasets and comparing to suitable base models

### Overall Weaknesses

* The variational inference approach here does not seem to be technically correct. The derived objective function and the update equations don't seem correct to me.
* Choice of experimental confounders needs more justification
* Experimental claims seem to lack significance testing, especially when comparing versions of the model with/without a component

**Additional Comments:**

# Presentation issues

Overall, sometimes a subscript indicates an integer index, other times it is a "decorator" like $W_u$ indicating this is for users (letter "u" for user). I found this double usage of subscripts confusing throughout. Perhaps you could use superscripts for decorative text and subscripts for indexing only? This is especially confusing for $\lambda_{R_{ij}}$ ... does it have a different value for each (i,j) pair?

#### on page 3

* hard to tell which variables depend on domain $m$ and which do not. What is size of $s$ relative to $s_m$? is $s$ a vector of M values, one for each domain?

* the generation of "v_j" item-specific embeddings and "x_j" item features do not need to be nested within the "for each user i" loop

* Notation choice of $\pi_i$ as an indicator from categorical distribution $\pi$ is not ideal... one is an integer, another is a vector of K probabilities

#### on page 5

in the "each term in Eqn. 8" list, the number 1 should be $E_q[ log p(u | \ldots)]$ not $E_q[ p (u | \ldots)]$

**Audience:**

Yes

**Audience Explanation:**

Recommender systems is a widely studied problem. I think this paper has an interesting take that others would appreciate, **provided technical issues above are resolved.**

**Broader Impact Concerns:**

No concerns.

**Claims And Evidence:**

No

**Claims Explanation:**

# Technical issues

### Provided ELBO equations missing crucial terms

The form of q(u) in Eq 6 is not described immediately. Is it assumed Gaussian?

The ELBO in Eq 8 is missing a needed entropy term for q(u). it only has one for q(v). Without this term, the ELBO objective is incomplete.

### Update for u/v in Eq 20-21 seems incorrect

The paper says they compute "the gradients of Eqn. 8 w.r.t. to u_i and v_j" above Eqn 20. However, Eqn. 8 is actually not a function of u_i or v_j. The ELBO is defined as an expectation over these random variables. So they are marginalized out. I don't see how these updates as written in eq 20-21 are valid.

Instead, following the usual VI recipe, one should derive updates for the parameters of $q(u)$ and $q(v)$. Perhaps these are "mean" u_i/v_j values and that's what's intended here, but it is not explained that way.

### Referring to the "E step" as updating W parameters in "Inference" paragraph is not correct

E step in EM/VI methods refers to an expectation step, where a *distribution* over variables of interest is updated. The point estimated parameters are instead estimated in an M step, as they are found via maximization of the objective.


### Strategy for categorical variable \pi and indicator \pi_i not clear

Is the categorical distribution $\pi$ estimated/learned? Fixed to uniform? I'd expect this to happen near the W updates in Alg 1

The estimation of \pi_i (cluster indicator for example i) does not seem clear in Sec. 2.4. I'm not clear why we need to condition on both R_ij and u_i... Looking at the PGM in Fig 2, \pi_i should be conditionally independent of R_ij given u_i.

### Causal confounder material in Sec 2.4 needs more development

We need more in the main paper about VDI than what is written in Sec. 2.4.  From these brief paragraphs, I don't yet appreciate why the do-calculus approach is needed.

# Experimental issues

### Choice of confounders needs more justification

I thought more setup explanation was needed in Sec. 3.1. Why is the production country a suitable confounder for the XMRec dataset? Why is the movie year a confounder for movies? Earlier in the intro, there was nice narrative about exposure bias (products featured in lots of ads are more likely to be bought) and popularity bias (products that are very popular are more likely to be bought). Is there any way to think about the year or the production country in this way?

### Some experimental claims seem to need significance testing

In Table 3, can we really believe that 0.1091 is significantly better than 0.1058? Or are they essentially equivalent? I'd recommend a bootstrap-based test for significant difference.

### Possible simple baseline missing

What if we just clustered the observable x_j features (or maybe observable per-user features) with a GMM (or other suitable mixture model), and then did per-cluster modeling to predict residuals? I guess I'm wondering if all this (admittedly cool) latent variable modeling is "worth it", or if there is some much easier procedure practitioners use as a wrapper to estimate residuals.

**Requested Changes:**

Without a significant rewrite of Sec. 2, I don't see myself able to vote for accepting this paper. The current VI treatment unfortunately has lots of issues, at least in how it is presented. I worry it may impact how the experiments were actually conducted. If so, I'd expect fresh experimental results using corrected inference/estimation code.

1) Rederive the VI methods (including the ELBO objective) accounting for $q(u)$ in addition to $q(v)$. Need to clearly define the assumed family of $q(u)$. Carefully rewrite the paper accordingly.

2) Either provide update equations for the parameters of $q(u)q(v)$ (rather than estimates of u/v), or explain why point estimating u/v makes sense when you've gone to the trouble of defining posteriors over these quantities.  In particular, it seems odd to not use the posterior mean of $v_j$ under q, and instead use a separate formula as in Eqn 21.

3) Adjust Alg 1 and surrounding text to clarify what is E-step and M-step better, and include updates for \pi

4) Improve narration about the causal confounder methods in Sec 2.4

5) Improve justification for choices of confounder in each experiment, ideally aligning to the reasons for the process in original Intro

---

> ### Author Response · Authors · 2025-11-14
> **Thank You for Your Encouraging and Construcive Comments [1/4]**
>
> Thank you for your encouraging and constructive comments. We are glad that you ``"like the modeling approach"`` and found our idea ``"interesting"`` / ``"have potential in this application"`` and our experiments ``"reasonable"``. Below, we address your questions one by one. We have revised the manuscript accordingly, with all changes marked in blue.
>
> ## Technical Questions
>
> **Q1: "The form of q(u) in Eq 6 is not described immediately. Is it assumed Gaussian?"**
>
> Yes, we assume $q(u)$ to be Gaussian, parameterized as: $q(\mathbf{u}\_i) = \mathcal{N}(\boldsymbol{\mu}\_{\mathbf{u}\_i}, \Lambda\_u^{-1}\mathbf{I})$.
>
>
> **Q2: "The ELBO in Eq 8 is missing a needed entropy term for q(u). it only has one for q(v)."**
>
> We apologize for this typo. You are correct that there is an entropy term for $u$, i.e., $-EB\_{q(u\_i,v\_j|x^v\_j)}\big[\log q(u\_i)]$. We have fixed this typo in the paper.
>
> **Q3: "Update for u/v in Eq 20-21 ... The paper says they compute "the gradients of Eqn. 8 w.r.t. to u_i and v_j" above Eqn 20. However, Eqn. 8 is actually not a function of u_i or v_j ..."**
>
> We are sorry for the confusion. We should have clarified that by "the gradients of Eqn. 8 w.r.t. to u_i and v_j" we mean the "the gradients of Eqn. 8 w.r.t. the means of q(u_i) and q(v_j)". Essentially, with the variational distributions with fixed (co)variance, learning q(u_i) and q(v_j) is equivalent to learning the means of these distributions.
>
> We have revised the paper accordingly to clarify this as suggested.
>
> **Q4: "Referring to the "E step" as updating W parameters in "Inference" paragraph is not correct"**
>
> This is a good point. We agree that updating $W$ parameters ($W_u$, $W_v$, and $w_R$) as well as $\mu_k$ and $\Sigma_k$ is the "M step" since (1) these are global parameters *shared across different users/items* and (2) they are treated as *deterministic* parameters rather than distributions. In this context, inferring latent variables like $u_i$ is treated as the E step.
>
> We have revised the paper accordingly as suggested.
>
> **Q5: "Is the categorical distribution estimated/learned? Fixed to uniform? I'd expect this to happen near the W updates in Alg 1"**
>
> Thank you for mentioning this. The categorical distribution $\pi_i$ is *estimated* rather than fixed to uniform.
>
> The estimation of $\pi$ is similar to that of a Gaussian mixture model (GMM) and we therefore omit it in the paper. Specifically, with a uniform prior distribution on $\pi_i$, i.e., $p(\pi_i=k|\theta)=\frac{1}{K}$, the posterior distribution $\pi_i$ is estimated as:
>
> $p(\pi_i = k | u_i, v_j, x_j^v, \\{\mu_k, \Sigma_k\\}^K_{k=1}) =\frac{\frac{1}{K}\mathcal{N}(u_i;\mu_k + W_u s_{m_i},\Sigma_k)}
> {\sum_{l=1}^K \frac{1}{K}\mathcal{N}(u_i;\mu_l + W_u s_{m_i},\Sigma_l)} = \frac{\mathcal{N}(u_i;\mu_k + W_u s_{m_i},\Sigma_k)}
> {\sum_{l=1}^K \mathcal{N}(u_i;\mu_l + W_u s_{m_i},\Sigma_l)}.
> $
>
>
> **Q6: "I'm not clear why we need to condition on both R_ij and u_i... Looking at the PGM in Fig 2, \pi_i should be conditionally independent of R_ij given u_i."**
>
> Yes, you are correct that $\pi_i$ is conditionally independent of $R$ given $u_i$. This is why we use $p(\pi_i = k | u_i, v_j, x_j^v, \{\mu_k, \Sigma_k\}_{i=1}^K) $ in the **"Summary"** paragraph before Section 3.
>
> **Q7: "We need more in the main paper about VDI than what is written in Sec. 2.4. From these brief paragraphs, I don't yet appreciate why the do-calculus approach is needed."**
>
> We are sorry for the confusion.
>
> **Why Do-Calculus Is Needed.** For example, in the dataset XMRec, each domain contains users and items from one country. "Country" (whose embedding is $s_m$ in Section 2.4) can be a confounder and creates biases similar to exposure bias and spurious correlation. For example, a model might learn that “users who like Bollywood films also like cricket gear,” not because of true shared interest, but because both are popular in India. To correct for such biases, we then need do-calculus to perform causal inference.
>
> **Domain Index $s_m$.** The domain index $s_m$ can be thought of as the embedding for each domain $m$. For example, in the dataset XMRec where each of the 18 domains contains items and users from one market/country (e.g., France or US), $s_m$ can be thought of as a "country" embedding. Interestingly, our preliminary results show that similar countries tend to have similar domain embedding $s_m$ (i.e., domain index in Xu et al., 2023). In other words, $s_m$ captures the similarities among different domains and therefore provide valuable information for our recommender systems.
>
> We have incorporated the discussion above in the revision as suggested.

---

> ### Author Response · Authors · 2025-11-14
> **Thank You for Your Encouraging and Construcive Comments [2/4]**
>
> ## Experimental Questions
> **Q8.1: "Why is the production country a suitable confounder for the XMRec dataset?"**
>
> This is a good question. In the dataset XMRec, each domain contains users and items from one country. "Country" (whose embedding is $s_m$ in Section 2.4) can be a confounder and creates biases similar to exposure bias and spurious correlation. For example, a model might learn that "users who like Bollywood films also like cricket gear," not because of true shared interest, but because both are popular in India. To correct for such biases, we then need do-calculus to perform causal inference and adjust for the confounder.
>
> **Q8.2: "Why is the movie year a confounder for movies? Earlier in the intro, there was nice narrative about exposure bias (products featured in lots of ads are more likely to be bought) and popularity bias (products that are very popular are more likely to be bought). Is there any way to think about the year or the production country in this way?**
>
> This is a good question. In the MovieLens dataset, movie year (whose embedding is $s_m$ in Section 2.4) can also be a confounder and creates biases similar to exposure bias and spurious correlation. This is because "movie year" influences both the movies which users are exposed to and the users’ observed preferences. When not properly controlled, "movie year" introduces temporal and popularity biases that distort the estimation of users’ true tastes and degrade the causal validity of recommendations.
>
> For example, below are some possible biases induced by "movie year":
>
> + **Availability Bias:** Users can only watch movies that have been released or are currently available (e.g., on Netflix or in theaters). Older movies are less likely to appear on the front page or in trending lists.
>
> + **Platform Curation:** Streaming services prioritize newly released or recently trending movies; thus, recent years correlate with higher exposure probabilities.
>
> + **Popularity Decay:** Movie visibility decreases over time, not necessarily because of quality, but due to algorithmic freshness or user attention dynamics.
>
> **Q9: "In Table 3, can we really believe that 0.1091 is significantly better than 0.1058?**
>
> We are sorry for the confusion. Note that both 0.1091 and 0.1058 are Recall@20; we should have emphasized that:
>
> (1) In recommender systems, mAP and NDCG are actually more important and commonly used metrics compared to the other metrics (i.e., Recall, F1, and Precision).
>   + For example, Recall@20, Precision@20, and F1@20 only measure the method's performance at **a single cutoff point, i.e., 20 recommended items**. In contrast, mAP@20 can be considered a variant of area under the precision-recall curve (AUPRC), thereby capturing the method's performance **across different numbers of recommended items from 1 to 20**. Therefore, mAP@20 is a much more comprehensive metric compared to Recall@20, Precision@20, and F1@20.
>   + Similarly, NDCG@20 assigns more weight to a correctly recommended item if it appears higher in the ranked list produced by the evaluated method. This offers more information than Recall@20, Precision@20, and F1@20, which treat all correct recommendations equally.
>
> (2) Note that both "PRUC (Full)" and "PRUC w/o Causality" are our proposed methods and count as our new contributions in the paper. Compared to existing methods,
> + "PRUC w/o Causality" proposes a new residual model under a probabilistic framework.
> + "PRUC (Full)" improves upon "PRUC w/o Causality" with causal inference to perform debiasing and adjust for confounders.
>
> (3) From Table 3, we can see that "PRUC (Full)" does have a clear improvement margin over "PRUC w/o Causality", therefore verifying the effectiveness of our causality component. Similarly, "PRUC w/o Causality" significantly outperforms the corresponding base model, demonstrating the effectiveness of our PRUC's probabilistic residual modeling.

---

> ### Author Response · Authors · 2025-11-14
> **Thank You for Your Encouraging and Construcive Comments [3/4]**
>
> **Q10: "What if we just clustered the observable x_j features (or maybe observable per-user features) with a GMM (or other suitable mixture model), and then did per-cluster modeling to predict residuals?"**
>
> This is a good question. Our preliminary experiments showed that such a simple baseline cannot achieve satisfactory performance. For example, Table A below shows some results comparing this simple baseline, i.e., "Clustering + Residuals", with our full PRUC, verifying such latent variable modeling is "worth it" as you mentioned.
>
>
>
> Table A. Results on the simple baseline and PRUC.
>
> | Method | Recall@20 | F1@20 | MAP@20 | NDCG@20 | Precision@20 |
> |---------|------------|--------|----------|-------------|---------------|
> | CDL (Base Model) | 0.0143 | 0.0016 | 0.0028 | 0.0009 | 0.0009 |
> | CDL (Clustering + Residuals) | 0.0156 | 0.0018 | 0.0023 | 0.0009 | 0.0010 |
> | PRUC (Full) | **0.1091** | **0.0128** | **0.0463** | **0.0108** | **0.0068** |
> ||
> | DLRM (Base Model) | 0.0044 | 0.0004 | 0.0004 | 0.0002 | 0.0002 |
> | DLRM (Clustering + Residuals) | 0.0163 | 0.0018 | 0.0029 | 0.0009 | 0.0010 |
> | PRUC (Full) | **0.0295** | **0.0035** | **0.0048** | **0.0018** | **0.0018** |
> ||
> | PerK (Base Model) | 0.1098 | 0.0128 | 0.0512 | 0.0112 | 0.0068 |
> | PerK (Clustering + Residuals) | 0.1118 | 0.0129 | 0.0513 | 0.0113 | 0.0069 |
> | PRUC (Full) | **0.1635** | **0.0192** | **0.0637** | **0.0151** | **0.0102** |
> ||
> | NCF (Base Model) | 0.0131 | 0.00148 | 0.0026 | 0.0008 | 0.0008 |
> | NCF (Clustering + Residuals) | 0.0164 | 0.0019 | 0.0029 | 0.0010 | 0.0010 |
> | PRUC (Full) | **0.1137** | **0.0137** | **0.0309** | **0.0090** | **0.0073** |
> ||
> | LightGCN (Base Model) | 0.0182 | 0.0021 | 0.0050 | 0.0014 | 0.0011 |
> | LightGCN (Clustering + Residuals) | 0.0277 | 0.0031 | 0.0054 | 0.0017 | 0.0017 |
> | PRUC (Full) | **0.1003** | **0.0121** | **0.0316** | **0.0084** | **0.0064** |
>
>
> We have incorporated these new results in the revision (Appendix B.4) in case future readers are also interested.
>
> ## Requested Changes
> **Q11: "I worry it may impact how the experiments were actually conducted."**
>
> Thank you for mentioning this. Actually we believe your comments can be easily addressed with our clarifications (see our response above) as most are simply due to either lack of details or typos. They therefore do not affect the implementation and experimental results.
>
> **Q12: "Rederive the VI methods (including the ELBO objective) accounting for $q(u)$ in addition to $q(v)$. Need to clearly define the assumed family of $q(u)$. Carefully rewrite the paper accordingly.""**
>
> Please see our **Response to Q1** above. We have also revised the paper accordingly.
>
> **Q13: "Either provide update equations for the parameters of $q(u)q(v)$(rather than estimates of u/v), or explain why point estimating u/v makes sense when you've gone to the trouble of dening posteriors over these quantities. In particular, it seems odd to not use the posterior mean of v_j under q, and instead use a separate formula as in Eqn 21."**
>
> Please see our **Response to Q2** above. We have also revised the paper accordingly.
>
> **Q14: "Adjust Alg 1 and surrounding text to clarify what is E-step and M-step better, and include updates for \pi."**
>
> Please see our **Response to Q3-Q5** above. We have also revised the paper accordingly.
>
> **Q15: "Improve narration about the causal confounder methods in Sec 2.4."**
>
> Please see our **Response to Q7** above. We have also revised the paper accordingly.
>
> **Q16: "Improve justication for choices of confounder in each experiment, ideally aligning to the reasons for the process in original Intro."**
>
> Please see our **Response to Q8.1, Q8.2, and Q9** above. We have also revised the paper accordingly.
> ## Presentation Problems
>
> **Q17: "Perhaps you could use superscripts for decorative text and subscripts for indexing only."**
>
> This is a good suggestion. We have revised our notations accordingly as suggested.
>
> **Q18: "This is especially confusing for $\lambda_{R_{ij}}$... does it have a different value for each (i,j) pair"**
>
> We are sorry for the confusion. Following CDL [1], during training: (1) for user-item pairs $(i, j)$ with **unobserved** ratings, we treat $R_{ij}$ as $0$ and set $\lambda_{R_{ij}}$ to $0.01$, indicating **lower confidence**, (2) for user-item pairs $(i, j)$ with **observed** ratings, we set $\lambda_{R_{ij}}$ to $1$, indicating **higher confidence**.
>
> [1] Wang, Hao, Naiyan Wang, and Dit-Yan Yeung. "Collaborative deep learning for recommender systems." KDD 2015.
>
> **Q19: "What is size of 's' relative to 's_m'? is 's' a vector of M values, one for each domain?"**
>
> We are sorry for the confusion. We should have made it clearer that we omit the subscript $m$ in $s_m$ when the context is clear. Therefore we use $s$ and $s_m$ interchangeably. $s_m$ is a $g$-dimensional vector where $m\in \{1, 2, \dots, M\}$ indexes one of the $M$ domains  (more details in Section 2.1). Therefore in total we have $M$ $s_m$'s, one for each domain.

---

> ### Author Response · Authors · 2025-11-14
> **Thank You for Your Encouraging and Construcive Comments [4/4]**
>
> **Q20: "the generation of "v_j" item-specificc embeddings and "x_j" item features do not need to be nested within the "for each user i" loop"**
>
> This is a good point. We have revised the paper (Section 2.2) accordingly.
>
> **Q21: "Notation choice of as an indicator from categorical distribution is not ideal... one is an integer, another is a vector of K probabilities"**
>
> We are sorry for the confusion. In the generative process in Section 2.2, we have changed "draw the user cluster ID $\pi_i$ from catagorical distribution $\pi$" to "draw the user cluster ID $\pi_i$ from a catagorical distribution $p(\pi_i | \theta)$" to avoid confusion and notation overload.
>
> **Q22: "on page 5 in the "each term in Eqn. 8" list, the number 1 should be $E_q[\log p(u \mid \ldots)]$ not $E_q[p(u \mid \ldots)]$"**
>
> Thank you for mentioning this. We have corrected the typo accordingly.
>
> Last but not least, we would like to thank Reviewer c93F for the careful review and constructive comments, especially on the typos and notations of the equations; they have greatly improved our manuscript. As shown in the revised manuscript, all concerns can be addressed through clarifications and fixing the typos, without affecting our experiments or our conclusion.
>
> Please do not hesitate to let us know if you have any follow-up questions, which we will be more than happy to address.

---

> > ### Comment · Reviewer_c93F · 2025-12-03
> > **Numerous correctness issues remain in the math foundations of the presented method**
> >
> > I do appreciate the authors' efforts to improve the mathematical clarity of the presented method.
> >
> > Unfortunately, even in revision the detailed math describing the method continues to contain numerous technical errors that impact a careful reader's ability to believe the overall method is correct as presented
> >
> > Here's a brief list from my review of the latest PDF:
> >
> > * I1: $\pi$ is a random variable in the graphical model (Fig. 2), yet it is conditioned on as a point estimate in the ELBO in Eq 9. The ELBO needs to include both a prior $p(\pi)$ and an approximate posterior $q(\pi)$, if the authors handle it as described in the response to Q5. I think it is confusing to "omit it from the paper" as the authors describe in their response. The current Eq 23 and surrounding text is also confusing, as it is disconnected from the ELBO presentation.
> >
> > * I2: instead of the current Eq 6, it should be made clear that the overall assumption is that $q( u, v ) = \prod_{i} q(u_i) \cdot \prod_j q(v_j | x_j)$. The current equation makes it seem like there is a separate posterior for each $i,j$ pair. A coherent statement about all approximate posteriors for all $i,j$ is needed.
> >
> > * I3: in Eq 8, it is ambiguous if the approximate posterior $q$ for $u_i$ has shared mean parameter $\mu$ across all values of $i$, or a different mean for each value of $i$
> >
> > * I4: below Eq 13, trying to define the "log likelihood for each cluster k", the left hand side is meant to describe a likelihood for one index $i$ as written, but the right hand side includes a sum over $i$ in the index set $I_k$
> >
> > * I5: Eq 16 is meant be be a KL between distributions over random variable $v_j$. Yet the left hand side assumes a specific value of $v_j$ is known. This is not correct, the KL should be a function of parameters of the model and inference distributions.
> >
> > * I6: what is hiding in the constant "C" term in various equations like 14-16 should be made more clear. For example, in Eq 16, the constant will depend on the values of the approximate posterior precision matrix $\Lambda_v$, so we cannot omit the constant when optimising for that parameter
> >
> > * I7: the current notation for the updates to u_i and v_j around Eq 21-22 in latest version is still unsatisfactory. I appreciate that the text has been updated to refer to the "means" of q(u_i) and q(v_j). However, it is not good practice to use $u_i$ to refer to the mean parameter in Eq 21 while simultaneously using the same symbol $u_i$ as a random variable elsewhere.
> >
> > * I8: The update to $q(u_i)$ in Eq 21 seems to rely on a deterministic value of $\pi_i$ (e.g. a specific class assigned to user i). However, there will be an approximate posterior $q(\pi)$ (something like the current Eq 23), not a deterministic discrete value for $\pi_i$. So this update should be written in terms of a soft assignment, not a hard assignment.
> >
> > Even if these specific items were fixed cosmetically in the next revised PDF submitted, given the patterns across the last two cycles of review unfortunately my confidence that overall the detailed technical implementation was handled correctly is not high.
> >
> > Given the severity of these correctness issues, I'll hold off on reviewing any experimental methods / results for now.

---

> > > ### Author Response · Authors · 2025-12-06
> > > **Thank You for Your Further Construcive Comments [2/2]**
> > >
> > > **"I6: what is hiding in the constant "C" term in various equations like 14-16 should be made more clear. For example, in Eq 16, the constant will depend on the values of the approximate posterior precision matrix $\Lambda_v$, so we cannot omit the constant when optimising for that parameter"**
> > >
> > > Actually the constant "C" can be omitted because we treat $\lambda_v \in \mathbb{R}$, $\lambda_u \in \mathbb{R}$, $\Lambda_v \in \mathbb{R}$, and $\Lambda_u \in \mathbb{R}$ as **constant scalars**.
> > >
> > > Note that slightly different from convention, as mentioned below Eq 8 and Eq 9, $\Lambda_v$ and $\Lambda_u$ are scalars, not matrices. Therefore the covariance matrices for $q(v_j)$ and $q(u_i)$ are $\Lambda_v^{-1} I $ and $\Lambda_u \Lambda_v^{-1} I$, respectively.
> > >
> > > We have clarified this below Eq 16 in the revision as suggested.
> > >
> > > **"I7: the current notation for the updates to u_i and v_j around Eq 21-22 in latest version is still unsatisfactory. I appreciate that the text has been updated to refer to the "means" of q(u_i) and q(v_j). However, it is not good practice to use $u_i$ to refer to the mean parameter in Eq 21 while simultaneously using the same symbol $u_i$ as a random variable elsewhere."**
> > >
> > > Throughout the original version of the paper, the covariance matrices for $q(v_j)$ and $q(u_i)$ are constant and isotropic (e.g., $\Lambda_v^{-1} I$ where $\Lambda_v$ is a constant scalar and $I$ is an identity matrix). Therefore updating the distribution $q(v_j)$ is equivalent to updating its mean. This is why we use $u_i$ to refer to both the random variable and the mean parameter, thereby minimizing notation and improving readability for non-Bayesian readers.
> > >
> > > We agree with the reviewer that this could cause confusion and have revised Eq 21-22 accordingly, e.g., changing $u_i$ to $\mu_{u_i}$.
> > >
> > > **"I8: The update to in Eq 21 seems to rely on a deterministic value of (e.g. a specific class assigned to user i). However, there will be an approximate posterior (something like the current Eq 23), not a deterministic discrete value for . So this update should be written in terms of a soft assignment, not a hard assignment."**
> > >
> > > Thank you for mentioning this. We understand where the reviewer comes from, and would like to clarify that we are using hard EM (i.e., the Classification EM, or CEM, as described in Eq 3.1-3.4 of [1]) rather than soft EM here because
> > > + Empirically, our preliminary results show that hard EM achieves comparable and sometimes even better performance compared to soft EM.
> > > + In production systems, it is often desirable to assign each user to only one user cluster to minimize computational cost, which is *proportional to the number of clusters used for each user*. In some sense, this is also similar to the popular mixture of experts (MOE) in large language models, where one often uses hard assignment in place of soft assignment.
> > >
> > > We hope the clarifications above are helpful and that the reviewer will also consider the empirical results presented in the original rebuttal. We have put considerable effort into obtaining these results.
> > >
> > > [1] A classification EM algorithm for clustering and two stochastic versions. Computational Statistics & Data Analysis 1992.

---

> > > > ### Comment · Reviewer_c93F · 2025-12-12
> > > > **Unfortunately substantial clarity and correctness questions remain**
> > > >
> > > > I appreciate the authors' efforts to revise the work further.
> > > >
> > > > Unfortunately, even in this second revision the detailed math describing the method continues to contain numerous communication issues and technical errors that impact my ability to verify that the overall method is correct and reproducible.
> > > >
> > > > * the new Eq 1 suddenly now uses a new odd "product of two Gaussians" construction, which I doubt most readers are familiar with and is likely to create confusion. It would be far better to just use the older formulation in past revision (dated Nov 14), that describes x and v as drawn from two Gaussians. There are several issues here in the new Eq 1:
> > > >
> > > > 	* The version now does not give any model for x, despite x being a random variable in the graphical model. This will puzzle any reader familiar with graphical models.
> > > > 	* It is now difficult to understand in Eq 1 what the roles of $\Lambda_v$ and $\lambda_v$ are (since they both use subscript v). These parameters are introduced to the reader without clear description of the roles they play or even their proper dimensions. Other quantities in Eq 1 are similarly ill-defined: how big is $v$ or the output of $f_v$?
> > > > 	* Eq 1 is also written as $p(v | x, s) \sim PoG(...)$ ... do you really mean that the density function is drawn from a distribution? Presumably there should be an "=" instead of $\sim$.
> > > >
> > > > * In Eq 8, the precision of the approximate posterior q(v_j) is defined as $\Lambda_v$, but that symbol was already used to define $p(v_j | ...)$ in the generative model (in Eq 6). Are these distict parameters, or purposefully coupled? The text is unclear.
> > > >
> > > > * RE I8 and the claim of "hard EM". Hard EM may be a valid way to approach this problem, but your paper is quite clear in Eq 23 and Alg 1 that you are computing *soft* not hard probabilities. So as written, the updates in Eq 21-22 (which assume a hard $\pi$) are ill-defined.
> > > >
> > > >
> > > > More minor but still confusing: In general, the paper is full of statements where there's a pdf function on the left hand side, but the value of the random variable isn't used on the right hand side (see for example Eq 4-6). It is further odd/confusing how Eq 4-6 essentially repeat verbatim earlier information given in the nested list describing the model near the top of this section.
> > > >
> > > > I know these comments may feel frustrating. I appreciate that your revisions are trying to improve the work and take substantial time. At the end of the day, my instructions as a reviewer are to answer the question: are the *claims made in the submission supported by accurate, convincing and clear evidence*? At the moment, I don't see a way I can answer yes to this question without a far more substantial overhaul of the math foundations of this paper.

---

> > > > > ### Author Response · Authors · 2025-12-13
> > > > > **Thank You for Your Further Comments [1/2]**
> > > > >
> > > > > Thank you again for your further constructive comments. Below we address your comments in detail one by one.
> > > > >
> > > > > **Q1.1: the new Eq 1 suddenly now uses a new odd "product of two Gaussians" construction, which I doubt most readers are familiar with and is likely to create confusion. It would be far better to just use the older formulation in past revision (dated Nov 14), that describes x and v as drawn from two Gaussians. There are several issues here in the new Eq 1:**
> > > > >
> > > > > We are sorry for the confusion. We agree that "Product of two Gaussians" is not commonly used, but it is not an odd construction. It is an established technique to model a random variable that is simultaneously regularized by two Gaussians. It has been used in variational learning of Gaussian belief networks by Geoffrey Hinton [1], speech recognition by Gales and Airey [2], and recommender systems [3]. The fact that it is not popular now does not make it any less useful.
> > > > >
> > > > > In the context of our paper, it describes the model where our item latent vector $v$ is simultaneously regularized (affected) by two Gaussians, one from the deep learning encoder, $\mathcal{N}(f_v(x_j^v), \Lambda_v^{-1}I)$ and one from the domain index, $\mathcal{N}(W^v s^{m}, \lambda_v^{-1}I)$. This matches our Eq. 16 in the paper, where one can see that the mean of $q(v_j)$, i.e., $\mu_{v_j}$, is regularized by both the projected domain index $W^v s_m$ and the encoding $f_v(x_j^v)$ from the deep learning encoder.
> > > > >
> > > > > Note that the "Product of two Gaussians" is also a Gaussian, i.e., $\mathcal{N}(\mu_{pog},\lambda_{pog}I)$, where
> > > > > $\mu_{pog} = \frac{\Lambda_v f_v(x_j^v) + \lambda_v W^v s^{m}}{\Lambda_v + \lambda_v}$ and $\lambda_{pog} = \Lambda_v + \lambda_v$ (more details in the revised paper around Eq. 1 and 2). Therefore effectively $v$ (or its mean $\mu_{v_j}$) is regularized towards a weighted average of $f_v(x_j^v)$ and $W^v s^{m}$, where $\Lambda_v$ and $\lambda_v$ are their corresponding weights, respectively (this is also related to your **Q1.3** on the roles of $\Lambda_v$ and $\lambda_v$ below).
> > > > >
> > > > >
> > > > >
> > > > > [1] Variational Learning in Nonlinear Gaussian Belief Networks. Neural Computation, 1999.
> > > > >
> > > > > [2] Product of Gaussians for Speech Recognition. Computer Speech & Language, 2006.
> > > > >
> > > > > [3] Collaborative Topic Regression with Social Regularization for Tag Recommendation, IJCAI, 2013.
> > > > >
> > > > >
> > > > > **"Q1.2: The version now does not give any model for x, despite x being a random variable in the graphical model. This will puzzle any reader familiar with graphical models."**
> > > > >
> > > > > Actually Figure 2 is still a valid graphical model because $x$ is an observed variable and is a root node. Note that in Figure 2, the arrow is from $x^v$ to $v$, i.e., $x^v \rightarrow v$, rather than $x^v \leftarrow v$. Here $x^v$ (i.e., the variable $x$ that you mentioned) serves as a conditioning variable in the conditional probability $p(v_j | x_j^v, s)$.
> > > > >
> > > > > **Q1.3: "It is now difficult to understand in Eq 1 what the roles of $\Lambda_v$ and $\lambda_v$ are (since they both use subscript v). These parameters are introduced to the reader without clear description of the roles they play or even their proper dimensions. Other quantities in Eq 1 are similarly ill-defined: how big is $v$ or the output of $f_v$?"**
> > > > >
> > > > > Sorry for the confusion. Actually $\Lambda_v$ and $\lambda_v$ have been in the paper since the original version, and the roles of $\Lambda_v$ and $\lambda_v$ remain unchanged compared to the original version.
> > > > >
> > > > > Specifically, our item latent vector $v$ is regularized by two Gaussians, one from the deep learning encoder, $\mathcal{N}(f_v(x_j^v), \Lambda_v^{-1} I)$ and one from the domain index, $\mathcal{N}(W^v s^{m}, \lambda_v^{-1} I)$. Therefore $\Lambda_v$ and $\lambda_v$ control the covariance matrices of these two Gaussians, respectively. This was also clarified in the text above Eq. 2 and in our **response to Q1.1** above.
> > > > >
> > > > > Practically speaking, this will affect which of the two Gaussians has a stronger regularization effect on the random variable $v_j$ (mostly its mean $\mu_{v_j}$). As shown in Eq. 16, $-\frac{\lambda\_v}{2} || \mu\_{v_j} - W^v s\_m ||^2 $ $-\frac{\Lambda\_v}{2} | \mu\_{v\_j} - f_v(x\_j^v) |^2$, a larger $\Lambda_v$ means $\mu_{v_j}$ is more affected by the deep learning encoding of $x_j^v$, i.e., $f_v(x_j^v)$.
> > > > >
> > > > > **Q1.4: "Eq 1 is also written as $p(v | x, s) \sim PoG(...)$ ... do you really mean that the density function is drawn from a distribution? Presumably there should be an "=" instead of $\sim$."**
> > > > >
> > > > > Yes, you are correct. We meant to say that $v\sim p(v | x, s) = PoG(...)$. We have fixed this in the revision to avoid confusion, as suggested.

---

> > > > > ### Author Response · Authors · 2025-12-13
> > > > > **Thank You for Your Further Comments [2/2]**
> > > > >
> > > > > **Q2: "In Eq 8, the precision of the approximate posterior q(v_j) is defined as $\Lambda_v$, but that symbol was already used to define $p(v_j | ...)$ in the generative model (in Eq 6). Are these distict parameters, or purposefully coupled? The text is unclear."**
> > > > >
> > > > > This is a good observation. Yes, they are purposefully coupled. We intentionally use the same precision $\Lambda_v$ for both $p(v_j | ...)$ and $q(v_j)$ for convenience and to reduce the number of hyperparameters. Intuitively it also makes sense to use the same precision for part of the PoG prior, i.e., $p(v_j | x_j^v, s)$, and the approximate posterior, i.e., $q(v_j)$.
> > > > >
> > > > > **Q3: "RE I8 and the claim of "hard EM". Hard EM may be a valid way to approach this problem, but your paper is quite clear in Eq 23 and Alg 1 that you are computing soft not hard probabilities. So as written, the updates in Eq 21-22 (which assume a hard $\pi$) are ill-defined."**
> > > > >
> > > > > We understand where the confusion comes from. The key is that after computing $q(\pi_i = k)$ using Eq. 23, there is a discretization step (or argmax step) in "hard EM" to choose the optimal (most probable) cluster for each user $i$, i.e., the cluster assignment for user $i$ is set to $\pi_i = \arg\max_k q(\pi_i = k)$. Here we can use the notation $\hat{\pi}_i$ in place of $\pi_i$ to be more rigorous if needed. We have clarified this in the revision (below Eq. 23) as suggested.
> > > > >
> > > > > **Q4.1: More minor but still confusing: In general, the paper is full of statements where there's a pdf function on the left hand side, but the value of the random variable isn't used on the right hand side (see for example Eq 4-6).**
> > > > >
> > > > > We agree that we could use more rigorous notation like $p(\tilde{R}\_{ij}|u\_{i}, v\_j, s_m)=\mathcal{N}(\tilde{R}\_{ij} | u\_{i}^T v_j+{w^R}^T s_{m}, \lambda\_{\tilde{R}\_{ij}}^{-1})$. However, this is more cumbersome and tedious notation since the random variable $\tilde{R}\_{ij}$ is clear from context. Therefore we follow the VDI paper (ICLR 2023) [4] (see Eq. 2$\sim$8 in [4] as examples) to omit it when the context is clear.
> > > > >
> > > > > [4] Domain-Indexing Variational Bayes: Interpretable Domain Index for Domain Adaptation. ICLR, 2023.
> > > > >
> > > > > **Q4.2: It is further odd/confusing how Eq 4-6 essentially repeat verbatim earlier information given in the nested list describing the model near the top of this section.**
> > > > >
> > > > > The nested list describing the model is typically referred to as the "generative process" in graphical model literature. It provides an overview of our model assumptions on how the observed variables are generated.
> > > > >
> > > > > Eq. 4-6 serve a different purpose and belong to the paragraph **"Model Factorization"**. Specifically, these equations
> > > > > + clarify how the joint distribution is factorized (i.e., Eq. 3) and
> > > > > + provide more details on each distribution. For example, below Eq. 4-6, the text clarifies that "$i$ and $j$ refers to the user index and the item index, respectively."
> > > > >
> > > > > This setup helps lay the groundwork for the introduction of different variational distributions as approximate posterior distributions (i.e., Eq. 7-9).
> > > > >
> > > > > That said, we are happy to remove them in the revision if you feel they are unnecessary.
> > > > >
> > > > > **Finally**, thank you again for keeping the communication channel open. We hope that through all these rounds of communication and clarification, the reviewer could understand why we chose to simplify the notations and equations and keep the description at a higher level in the first place. There are a lot of subtle details that only researchers in the Bayesian (and graphical models) community care about, and dumping them all in the main paper could easily lead to confusion, especially for general readers without a Bayesian background.
> > > > >
> > > > > We believe that after incorporating your comments, the revised paper is clearer with more details for a reader in the Bayesian community, and we are immensely grateful for your time. All implementations and experiments are still valid and the conclusions of this paper remain unchanged.

---

> ### Author Response · Authors · 2025-12-06
> **Thank You for Your Further Construcive Comments [1/2]**
>
> Thank you again for your further constructive comments. We would like to emphasize that we fully understand and appreciate the reviewer's rigor in the notations, etc. In many cases, we chose the notations to balance completeness (e.g., minimizing ambiguity and conforming to conventions in Bayesian inference literature) and readability (e.g., minimizing the use of subscripts, superscripts, and levels of summation $\sum$ and products $\prod$ in equations), as readers who are not from the Bayesian community may find some of the simplified notations more accessible. Below we address your comments one by one. We have also revised the manuscript accordingly, with the changes marked in blue.
>
> **"I1: $\pi$ is a random variable in the graphical model (Fig. 2), yet it is conditioned on as a point estimate in the ELBO in Eq 9. The ELBO needs to include both a prior $p(\pi)$ and an approximate posterior q(\pi), if the authors handle it as described in the response to Q5. I think it is confusing to "omit it from the paper" as the authors describe in their response. The current Eq 23 and surrounding text is also confusing, as it is disconnected from the ELBO presentation."**
>
> We are sorry for the confusion. As mentioned earlier, our intention was to simplify the presentation and improve readability for non-Bayesian readers. We therefore systematically move $\pi$ from the left of the conditional probability to the right of it so that we can focus on describing the inference of $u$ and $v$.
>
> You are correct that the complete version does include an additional term $KL(q(\pi) | p(\pi))$ in the ELBO, where $p(\pi)$ is a uniform prior as we mentioned earlier in the response to Q5 and around Eq 23. In fact, Eq 23 is derived exactly by maximizing this ELBO w.r.t. $\pi$. Therefore the equation is correct. $q(\pi_i=k)$ is exactly $p(\pi_i = k | u_i, v_j, x_j^v, \\{\mu_k, \Sigma_k\\}^K_{k=1})$ in Eq 23. We have revised the paper accordingly to clarify this.
>
> **"I2: instead of the current Eq 6, it should be made clear that the overall assumption is that $q(u,v)=\prod_i q(u_i) \prod_j q(v_j)$. The current equation makes it seem like there is a separate posterior for each $i,j$ pair. A coherent statement about all approximate posteriors for all is needed."**
>
> Thank you for your suggestion. We would like to clarify that the reason we use $q(u_i,v_j)= q(u_i) q(v_j)$ is so that it can be **consistent with Equations 1~5**, which describe the factorization for **one single $i,j$ pair**. This is part of our effort to improve readability for non-Bayesian readers (e.g., minimizing levels of summation $\sum$ and products $\prod$ over $i$ and $j$ in equations).
>
> We agree that this might introduce ambiguity, making it seem like there is a separate (approximate) posterior for each $i,j$ pair. Therefore following your suggestion, we have now revised Equation 6 in the manuscript accordingly to avoid confusion.
>
> **"I3: in Eq 8, it is ambiguous if the approximate posterior $q$ for $u_i$ has shared mean parameter $\mu$ across all values of $i$, or a different mean for each value of $i$"**
>
> In the approximate posterior, there is **a different mean** $\mu_{u_i}$ for each value of $i$, since different users should have different latent vectors.
> + In terms of notation, the subscript $i$ in $\mu_{u_i}$ indicates that this mean varies with $i$, and therefore $\mu_{u_i}$ differs across users.
> + The reviewer might confuse the approximate posterior with the prior. Note that while different users $i$ share the same "prior" if they are from the same cluster, their approximate posteriors are different.
>
> We have further clarified this in the revision (after Eq 8 and 9) in case other readers have similar confusion.
>
> **"I4: below Eq 13, trying to define the "log likelihood for each cluster k", the left hand side is meant to describe a likelihood for one index $i$ as written, but the right hand side includes a sum over $i$ in the index set $I_k$"**
>
> We are sorry for the typo. $u_i$ should be $\\{u_i\\}_{i \in I_k}$ since this equation is for all users inside cluster $I_k$. We have fixed this typo in the revision.
>
> **"I5: Eq 16 is meant be be a KL between distributions over random variable $v_j$. Yet the left hand side assumes a specific value of $v_j$ is known. This is not correct, the KL should be a function of parameters of the model and inference distributions."**
>
> This is a good point. As mentioned earlier, $v_j$ is a random variable. For example, $q(v_j)$ is a Gaussian distribution, $\mathcal{N}(\mu_{u_i}, \Lambda_u^{-1} I)$. Therefore all $v_j$'s should be replaced with $\mu_{u_i}$. We have revised the paper accordingly.

---

### Review · Reviewer_KV4b · 2025-10-30

**Summary Of Contributions:**

This paper introduces a post-processing algorithm called Probabilistic Residual User Clustering (PRUC), which is inspired by Bayesian causal inference. The algorithm is designed to enhance existing recommender systems by reducing confounding factor bias and improving generalization to out-of-sample data. PRUC achieves this by calibrating the recommender system through the modeling of scoring residuals, denoted as $\widetilde{R}$. It leverages the sub-structure within users by applying a mixture of Gaussian distributions to the user latent representations. A confounding factor, $S_m$, is then used to determine the distribution of user and item latent representations, which in turn influences the distribution of the residuals. The entire likelihood function is optimized using the ELBOW. During the inference stage, the rating residual is sampled from the estimated Gaussian distribution and undergoes a debiasing process controlled by the confounding factor $S_m$, ultimately adjusting the original rating prediction.

Multiple domain-transfer experiments were conducted on two public datasets, and ablation studies on certain variants of the proposed algorithm were performed to demonstrate the contribution of individual modules.

**Additional Comments:**

Not relevant

**Audience:**

Yes

**Audience Explanation:**

While there are still some concerns regarding the results and claims presented in this paper, it introduces a calibration procedure that can seamlessly integrate with many existing recommender system algorithms in a plug-and-play manner. The algorithm effectively highlights challenges that current recommender systems may face in domain-shifting scenarios. By approaching this issue from a Bayesian causal inference perspective, the algorithm offers a novel solution. This work could be of significant interest to the TMLR community, provided that many of the concerns are successfully addressed.

**Broader Impact Concerns:**

Not relevant

**Claims And Evidence:**

No

**Claims Explanation:**

While this paper makes several valuable contributions, as previously mentioned, I have some concerns regarding the methodology description, details of the experiments, and the ablation studies.

Concern 1: Although I understand that the experiment is designed to create a scenario where debiasing is essential for enhancing generalization performance, I believe the author could provide more analysis on the residuals to reinforce this point. For instance, conducting statistical analyses to demonstrate preference biases across different countries could be beneficial. It would be insightful to show how individuals from Country A might favor certain genres over others, and then illustrate how the proposed algorithm effectively debiases these preferences, thereby enabling the recommendations to generalize across different domains.

Additionally, it appears that some ablation studies are missing, which could further support the findings.

Concern 2: It would be helpful if the author could address the possibility that the observed performance improvement might be attributed to additional parameters. For example, if the number of layers or the width of the original recommender system were increased to match the parameter count of the PRUC-enriched network, would the performance remain comparable?

Concern 3: Another point to consider is whether the performance enhancement is due to the cascading effect rather than debiasing. For example, if a Neural Collaborative Filtering (NCF) model were used to fit the residuals and adjust the final recommendation score, would similar results be achieved?  If Figure 4 is presented for this purpose, please elaborate more on the w/o causality variant.

Concern 4: In Section 2.4, the author mentions that the causal confounder (s_m) is isolated using VDI by approximating its posterior (p(s_m | ...)). Could the author clarify if this implies that the confounder is the latent global indexing? However, in the experimental phase, it is stated that "we use the production country of the products as the causal confounders...". Further clarification on the isolation of the confounder would be appreciated.

Concern 5: If the production country of the products is used as a causal confounder, it seems to contradict the PGM model where the confounder (s_m) is drawn from a standard Gaussian prior. Since the production country, being a categorical variable, should not have a Gaussian as an uninformative prior, more elaboration on this point would be beneficial.

Concern 6:  I understand that the cold-start setting is designed to create a more challenging scenario, allowing the advantages of the proposed algorithm to be more clearly demonstrated. However, it would be beneficial to conduct ablation studies on datasets where a higher number of testing domain user records are present in the training set. This would provide a better understanding of the conditions under which the PRUC algorithm should be utilized.

**Requested Changes:**

As I mentioned above:

1. It would be beneficial to provide a more detailed analysis of the residuals. The author might consider visualizing the bias patterns with respect to the confounder and demonstrating how PRUC effectively addresses this bias.

2. An ablation study should be included where both the original recommender and the PRUC-enriched recommender operate with similar parameter sizes or FLOPs levels. This would help in understanding the impact of the algorithm under comparable conditions.

3. Please consider adding an ablation study where the residual is fitted using a Neural Collaborative Filtering (NCF) model instead of the proposed model. If such a study has already been conducted, it would be helpful to elaborate further on the different ablation variant models used.

4. More explanation is needed regarding the choice of confounders, as well as their isolation and prior assumptions. Please refer to the concerns I raised in the section titled "Explain Your Answer Above" for further context.

5. Conducting an ablation study in a less challenging scenario would be valuable. This would allow readers to understand the specific conditions under which the proposed method excels.

---

> ### Author Response · Authors · 2025-11-14
> **Thank You for Your Encouraging and Construcive Comments [1/5]**
>
> Thank you for your encouraging and constructive comments and for acknowledging that ``"this paper makes several valuable contributions"`` and that our experiments ``"demonstrate the contribution of individual modules"``. Below, we address your questions one by one. We have revised the manuscript accordingly, with all changes marked in blue.
>
>
> **Q1.1: It would be insightful to show how individuals from Country A might favor certain genres over others, and then illustrate how the proposed algorithm effectively debiases these preferences, thereby enabling the recommendations to generalize across different domains.**
>
> Thank you for this valuable suggestion. Following your suggestion and to demonstrate PRUC's debiasing capability and cross-domain generalization, we conducted a detailed analysis using the first domain pair of the XMRec dataset (in Table 1 of the paper), which corresponds to the countries **France, Italy, India, Japan, and Mexico**.
>
> We first examined the top-20 recommendations for each user using the CDL baseline model. The resulting distributions showed notable country-specific biases:
>
>   + Italian users: 34 "Camera & Photo" recommendations (3.70%) out of 920 total recommendations.
>   + Indian users: 878 "Camera & Photo" recommendations (6.46%) out of 13,600 total recommendations.
>
>   The bias ratio across countries was 1.75$\times$ (maximum: 6.46%, minimum: 3.70%), indicating that Indian users were recommended camera products 1.75 times more frequently than Italian users.
>
>   After applying our PRUC model, we examined the top-20 recommendations for each user. The resulting distributions were notably more balanced:
>
>   + Italian users: 33 "Camera & Photo" recommendations (3.59%) out of 920 total recommendations.
>   + Indian users: 427 "Camera & Photo" recommendations (3.14%) out of 13,600 total recommendations.
>
>   The bias ratio across countries was reduced to 1.14$\times$ (maximum: 3.59%, minimum: 3.14%), representing a 38.5% reduction in country-specific bias. This demonstrates that PRUC successfully mitigated the preference biases through probabilistic user clustering and causal debiasing, thereby enabling the model to learn more generalizable user-item interaction patterns.
>
> We have incorporated the discussion above in the revised paper (Section 3.4) as suggested.

---

> ### Author Response · Authors · 2025-11-14
> **Thank You for Your Encouraging and Construcive Comments [2/5]**
>
> **Q1.2: Additionally, it appears that some ablation studies are missing, which could further support the findings.**
>
> Thank you for mentioning this.
>
> **Ablation Studies in Figure 4, Table 3, and Table 4.** As also mentioned in the **Response to Q3** below, Figure 4, along with Table 3 and Table 4, does serve as important ablation studies, showing that the performance improvement is not only due to the cascading effect but also debiasing. Specifically, in Figure 4, "PRUC w/o Causality" is an ablated version of PRUC without the causal debiasing component while "PRUC (Full)" is the full version of PRUC with the causal debiasing component. Comparing "PRUC w/o causality"  and "PRUC (Full)", we can see that "PRUC (Full)" does outperform "PRUC w/o causality", verifying the effectiveness of the causal debiasing component in PRUC.
>
> **Additional Ablation Studies in the Appendix.** We also had additional ablation studies in Table 5$\sim$14 of the Appendix.
>
> **New Ablation Studies.** Following your suggestion, we have conducted additional ablation experiments to further validate our findings. For example, Table C below shows some results comparing this simple baseline, i.e., "Clustering + Residuals", with our full PRUC, verifying such latent variable modeling is important.
>
>
> Table C. Results on the simple baseline and PRUC.
>
> | Method | Recall@20 | F1@20 | MAP@20 | NDCG@20 | Precision@20 |
> |---------|------------|--------|----------|-------------|---------------|
> | CDL (Base Model) | 0.0143 | 0.0016 | 0.0028 | 0.0009 | 0.0009 |
> | CDL (Clustering + Residuals) | 0.0156 | 0.0018 | 0.0023 | 0.0009 | 0.0010 |
> | PRUC (Full) | **0.1091** | **0.0128** | **0.0463** | **0.0108** | **0.0068** |
> ||
> | DLRM (Base Model) | 0.0044 | 0.0004 | 0.0004 | 0.0002 | 0.0002 |
> | DLRM (Clustering + Residuals) | 0.0163 | 0.0018 | 0.0029 | 0.0009 | 0.0010 |
> | PRUC (Full) | **0.0295** | **0.0035** | **0.0048** | **0.0018** | **0.0018** |
> ||
> | PerK (Base Model) | 0.1098 | 0.0128 | 0.0512 | 0.0112 | 0.0068 |
> | PerK (Clustering + Residuals) | 0.1118 | 0.0129 | 0.0513 | 0.0113 | 0.0069 |
> | PRUC (Full) | **0.1635** | **0.0192** | **0.0637** | **0.0151** | **0.0102** |
> ||
> | NCF (Base Model) | 0.0131 | 0.00148 | 0.0026 | 0.0008 | 0.0008 |
> | NCF (Clustering + Residuals) | 0.0164 | 0.0019 | 0.0029 | 0.0010 | 0.0010 |
> | PRUC (Full) | **0.1137** | **0.0137** | **0.0309** | **0.0090** | **0.0073** |
> ||
> | LightGCN (Base Model) | 0.0182 | 0.0021 | 0.0050 | 0.0014 | 0.0011 |
> | LightGCN (Clustering + Residuals) | 0.0277 | 0.0031 | 0.0054 | 0.0017 | 0.0017 |
> | PRUC (Full) | **0.1003** | **0.0121** | **0.0316** | **0.0084** | **0.0064** |
>
>
> We evaluated PRUC without the regularization term to isolate the effect of the proposed regularization. In our **Response to Q2** below, we compared PRUC with larger CDL variants to control for model capacity and parameter count. Moreover, in our **Response to Q7** below, we performed n-shot experiments on both the CDL base model and PRUC, analyzing how performance changes with varying amounts of testing-domain user records included during training.
>
> Overall, these ablation results consistently demonstrate that the improvements of PRUC are not simply due to increased model complexity or data overlap, but are primarily attributed to its probabilistic and causal modeling mechanism, which effectively enhances knowledge transfer across domains.
>
> We will clarify our existing ablation studies in the paper and include new results above in the revision as suggested.

---

> ### Author Response · Authors · 2025-11-14
> **Thank You for Your Encouraging and Construcive Comments [3/5]**
>
> **Q2: It would be helpful if the author could address the possibility that the observed performance improvement might be attributed to additional parameters. For example, if the number of layers or the width of the original recommender system were increased to match the parameter count of the PRUC-enriched network, would the performance remain comparable?**
>
> This is a good point. Following your suggestion, to check whether the observed performance gain of PRUC is simply due to an increased number of parameters, we conducted an additional controlled experiment by scaling up the CDL baseline to approximately match the parameter size of the PRUC-enriched network. This experiment was performed on the first domain pair of the XMRec dataset (as shown in Table 1 of the paper), which includes data from **France, Italy, India, Japan, and Mexico**.
>
> Concretely, we expanded both the *depth* and *width* of the CDL architecture as suggested. The original CDL structure was 512 → 200 → 50, while the larger (deeper and wider) version used an architecture of 512 → 550 → 400 → 50, resulting in roughly **1.05M** parameters, which is comparable to PRUC’s **0.90M** parameters. This ensures that the comparison isolates the effect of PRUC from mere model capacity differences.
>
> The results are presented in Table D below. Although the deeper CDL exhibits a slight performance improvement over the base model, PRUC still achieves substantially higher performance across all metrics. This demonstrates that PRUC’s gains stem from its ability to capture cross-domain relational patterns, user-cluster-specific representations, and debiasing, rather than simply from an increased parameter count.
>
> Table D: Performance comparison between larger CDL and PRUC
>
> | Method | Recall@20 | F1@20 | MAP@20 | NDCG@20 | Precision@20 |
> |---------|------------|--------|----------|-------------|---------------|
> | CDL (Base Model) | 0.0143 | 0.0016 | 0.0028 | 0.0009 | 0.0009 |
> | CDL (Larger) | 0.0223 | 0.0026 | 0.0022 | 0.0011 | 0.0014 |
> | PRUC (Full) | **0.1091** | **0.0128** | **0.0463** | **0.0108** | **0.0068** |
>
> We have included the discussion and results above in the revised paper (Appendix B.4) as suggested.
>
>
> **Q3: Another point to consider is whether the performance enhancement is due to the cascading effect rather than debiasing. For example, if a Neural Collaborative Filtering (NCF) model were used to fit the residuals and adjust the final recommendation score, would similar results be achieved? If Figure 4 is presented for this purpose, please elaborate more on the w/o causality variant.**
>
> Thank you for mentioning this. Indeed, Figure 4 does show that the performance improvement is not only due to the cascading effect but also debiasing. Specifically, in Figure 4, "PRUC w/o Causality" is an ablated version of PRUC without the causal debiasing component while "PRUC (Full)" is the full version of PRUC with the causal debiasing component. Comparing "PRUC w/o causality"  and "PRUC (Full)", we can see that "PRUC (Full)" does outperform "PRUC w/o causality", verifying the effectiveness of the causal debiasing component in PRUC.
>
> **Q4: Could the author clarify if this implies that the confounder is the latent global indexing? ... "we use the production country of the products as the causal confounders..."**
>
> This is a good question. Yes, the confounder $s_m$ is the latent global indexing in VDI. Below we would like to clarify how $s_m$ is connected to "country".
>
> For example, in the dataset XMRec, for each user/item we have the "country" information. However, "country" here is simply a one-hot vector and does not contain semantics such as which countries are more similar to each other (e.g., US and Canada) and which countries are more different from each other (e.g., US and France).
>
> In contrast, the domain index $s_m$ can be thought of as the embedding for each domain $m$ (e.g., for each country). For example, in the dataset XMRec where each of the 18 domains contains items and users from one market/country (e.g., France or US), **$s_m$ can be thought of as a "country" embedding**. Interestingly, our preliminary results show that similar countries tend to have similar domain embedding $s_m$ (i.e., domain index in Xu et al., 2023). In other words, $s_m$ captures the similarities among different domains (e.g., different countries) and therefore provides valuable information for our recommender systems.

---

> ### Author Response · Authors · 2025-11-14
> **Thank You for Your Encouraging and Construcive Comments [4/5]**
>
> **Q5: If the production country of the products is used as a causal confounder, it seems to contradict the PGM model where the confounder (s_m) is drawn from a standard Gaussian prior.**
>
> We are sorry for the confusion. $s_m$ is actually a continuous embedding vector, not a categorical variable, as mentioned in the **Response to Q4** above.
>
> For example, in the dataset XMRec, for each user/item we have the "country" information. However, "country" here is simply a one-hot vector and does not contain semantics such as which countries are more similar to each other (e.g., US and Canada) and which countries are more different from each other (e.g., US and France). A simple example of one-hot vectors is $[1, 0, 0]$, $[0, 1, 0]$, and $[0, 0, 1]$ representing $3$ countries. We can see that these vector contrain limited information because the Euclidean distance bewteen any pair of one-hot vectors is identical, i.e., $1$.
>
> Therefore we follow [Xu et al., 2023] to compute the domain index $s_m$, which can be thought of as the embedding for each domain $m$ (e.g., for each country). For example, in the dataset XMRec where each of the 18 domains contains items and users from one market/country (e.g., France or US), $s_m$ can be thought of as a "country" embedding. Interestingly, our preliminary results show that similar countries tend to have similar domain embedding $s_m$ (i.e., domain index in Xu et al., 2023). In other words, $s_m$ is a continuous embedding vector that captures the similarities among different domains and therefore provides valuable information for our recommender systems.
>
> Therefore it is natural to use a standard Gaussian prior for $s_m$.

---

> ### Author Response · Authors · 2025-11-14
> **Thank You for Your Encouraging and Construcive Comments [5/5]**
>
> **Q6: However, it would be beneficial to conduct ablation studies on datasets where a higher number of testing domain user records are present in the training set.**
>
> This is a good suggestion. Following your advice, we conducted additional ablation experiments using CDL as the base model to analyze the effect of incorporating a higher number of testing-domain user records into the training process. Specifically, we used the second domain pair of the XMRec dataset (as presented in Table 1 of the paper), involving users from **Mexico, Spain, India, Japan, and Germany**.
>
> To control for user activity, we selected users with more than four interactions and varied the number of testing-domain user records included in the training set to construct **1-shot, 2-shot, and 3-shot** scenarios, where $n$-shot means there are $n$ records for each testing user in the training set. The performance of both the base CDL model and our proposed PRUC model in these settings is summarized in Table E and F below.
>
> Table E: Results in $n$-shot settings for the CDL base model
>
> ----------------------------------------------------------------------------------------
> |   Method   | Config     | Recall@20   | Precision@20  | F1@20       | MAP@20      | NDCG@20
> |-------------|------------|--------|----------|-------------|---------------|---------------|
> |   CDL (Base Model)  |   1-shot     | 0.0647   | **0.0105**     | 0.0181   | 0.0111   | 0.0098
> |   CDL (Base Model)  |   2-shot     | 0.0700   | 0.0085     | **0.0152**   | 0.0108   | 0.0078
> |   CDL (Base Model)  |   3-shot     | **0.0817**   | 0.0060     | 0.0112   | **0.0159**   | **0.0106**
>
> Table F: Results in $n$-shot settings for our proposed model PRUC (Full) using the CDL as the base model
>
> ----------------------------------------------------------------------------------------
> |   Method   | Config     | Recall@20   | Precision@20  | F1@20       | MAP@20      | NDCG@20
> |------------|-----------|---------|----------|------------|---------------|---------------|
> |   PRUC (Full)      |  1-shot      | 0.1080   | **0.0195**     | **0.0359**   | 0.0252   | 0.0208
> |   PRUC (Full)      |  2-shot      | 0.1507   | **0.0195**     | 0.0356   | 0.0359   | 0.0216
> |   PRUC (Full)      |  3-shot      | **0.2178**   | 0.0145     | 0.0272   | **0.0471**   | **0.0251**
>
> As shown in the tables, both CDL and PRUC benefit from the inclusion of additional testing-domain user records, confirming that more exposure to testing-domain data can improve cross-domain adaptation. More importantly, PRUC consistently outperforms CDL across all $n$-shot settings, demonstrating stronger few-shot generalization and more effective utilization of cross-domain information. These findings further validate the robustness and adaptability of PRUC when the overlap between training and testing domains increases.
>
> We have included the discussion and results above in the revised paper (Appendix B.4) as suggested.
>
> Last but not least, we would like to thank Reviewer KV4b for the careful review and constructive comments, especially on the suggestion of further ablation studies; they have further highlighted the effectiveness of our method and the necessity of probabilistic modeling.
>
> We believe all concerns have now been addressed. Please do not hesitate to let us know if you have any follow-up questions, which we will be more than happy to address.

---

### Author Response · Authors · 2026-01-01
**Final Clarification**

Dear Action Editor,

Thank you for overseeing the review of our manuscript.

We would like to clarify that in the **fourth** round of revision, we carefully addressed **all** previously raised questions regarding the Bayesian formulation and revised the manuscript accordingly. Unfortunately, after this revision, the reviewer who had raised these concerns **did not provide further feedback**, leaving no opportunity to respond to the updated presentation.

We appreciate the rigor of the review process and, in light of the decision, plan to resubmit the manuscript with a major revision.

Thank you for your time and consideration.

Warm regards,

Authors of PRUC

---

### Decision · Action_Editor_26Fz · 2025-12-28

**Recommendation:** Reject

**Audience:**

Yes

**Audience Explanation:**

Modern deep learning recommender systems often function as black boxes, frequently struggling with interpretability and performance degradation during domain shifts or cold-start scenarios. This paper proposes Probabilistic Residual User Clustering (PRUC), a plug-and-play causal Bayesian framework designed to enhance existing base recommenders. Rather than predicting ratings from scratch, PRUC focuses on residual rating prediction, which is a compelling approach. The causal debiasing component is both intuitive and effective, and the overall core concept is quite promising. However, while the methodology is conceptually sound, the paper (even in the second round of revision)  suffers from significant deficiencies in clarity and mathematical rigor that must be thoroughly addressed.

**Claims And Evidence:**

No

**Claims Explanation:**

Two of reviewers have concerns that even in the second round of revision, substantial issues with clarity and correctness exist in the math presentation in the paper. There seems to be flaws in a few different places (eqs. 4, 22, 24-26) in the algorithm derivation.

**Resubmission Of Major Revision:**

The authors may consider submitting a major revision at a later time.